

# Modelling wintertime Arctic Haze and sea-spray aerosols

Eleftherios Ioannidis[1], Kathy S. Law[1], Jean-Christophe Raut[1], Louis Marelle[1], Tatsuo Onishi[1], Rachel M. Kirpes[2], Lucia Upchurch[3], Andreas Massling[4], Henrik Skov[4], Patricia K. Quinn[3], and Kerri A. Pratt[2,5]

[1]LATMOS/IPSL, Sorbonne Université, UVSQ, CNRS, Paris, France
[2]Department of Chemistry, University of Michigan, Ann Arbor, Michigan, USA
[3]Pacific Marine Environmental Laboratory, National Oceanic and Atmospheric Administration, Seattle, Washington 98115, United States
[4]Department of Environmental Science, iClimate, Aarhus University, Denmark
[5]Department of Earth and Environmental Sciences, University of Michigan, Ann Arbor, Michigan, USA

**Correspondence:** Eleftherios Ioannidis (eleftherios.ioannidis@latmos.ipsl.fr) and Kathy S. Law (Kathy.Law@latmos.ipsl.fr)

**Abstract.** Anthropogenic and natural emissions contribute to enhanced concentrations of aerosols, so-called Arctic Haze in the Arctic winter and early spring. Models still have difficulties reproducing available observations. Whilst most attention has focused on the contribution of anthropogenic aerosols, there has been less focus on natural components such as sea-spray aerosols (SSA), including sea-salt sulphate and marine organics, which can make an important contribution to fine and coarse

mode aerosols, particularly in coastal areas. Models tend to underestimate sub-micron and overestimate super-micron SSA in polar regions, including in the Arctic region. Quasi-hemispheric runs of the Weather Research Forecast model, coupled with chemistry model (WRF-Chem) are compared to aerosol composition data at remote Arctic sites to evaluate the model performance simulating wintertime Arctic Haze. Results show that the model overestimates sea-salt (sodium and chloride) and nitrate and underestimates sulphate aerosols. Inclusion of more recent wind-speed and sea-surface temperature dependencies

for sea-salt emissions, as well as inclusion of marine organic and sea-salt sulphate aerosol emissions leads to better agreement with the observations during wintertime. The model captures better the contribution of SSA to total mass for different aerosol modes, ranging from 20-93% in the observations. The sensitivity of modelled SSA to processes influencing SSA production are examined in regional runs over northern Alaska (United States) where the model underestimates episodes of high SSA, particularly in the sub-micron, that were observed in winter 2014 during field campaigns at the Barrow Observatory, Utqiaġvik.

A local source of marine organics is also included following previous studies showing evidence for an important contribution from marine emissions. Model results show relatively small sensitivity to aerosol dry removal with more sensitivity (improved biases) to using a higher wind speed dependence based on sub-micron data reported from an Arctic cruise. Sea-ice fraction, including sources from open leads, is shown to be a more important factor controlling modelled super-micron SSA than sub-micron SSA. The findings of this study support analysis of the field campaign data pointing out that open leads are the primary

source of SSA, including marine organic aerosols during wintertime at the Barrow Observatory, Utqiaġvik. Nevertheless, episodes of high observed SSA are still underestimated by the model at this site, possibly due to missing sources such as SSA production from breaking waves. An analysis of the observations and model results does not suggest an influence from blowing snow and frost flowers to SSA during the period of interest. Reasons for the high concentrations of sub-micron SSA observed at this site, higher than other Arctic sites, require further investigation.





## 1 Introduction

The Arctic region is warming faster than any other region on Earth *(Allan, 2021)*. Greenhouse gases, in particular carbon dioxide, and short-lived climate forcers like methane, ozone and, aerosols have a significant impact on the environment, with a particularly strong warming effect in the Arctic region *(AMAP, 2015; Allan, 2021)*. This region is influenced by enhanced concentrations of aerosols (including sulphate ($SO_4^{2-}$), nitrate ($NO_3^-$), black carbon (BC) and organic aerosols (OA) during winter and spring, a phenomenon called Arctic Haze *(Rahn and McCaffrey, 1980; Barrie et al., 1994; Quinn et al., 2002)*. Transport of aerosols and their precursors from mid-latitudes anthropogenic emissions contribute to Arctic Haze *(Heidam et al., 2004; Quinn et al., 2007; Law et al., 2014)*. Local within and near-Arctic anthropogenic and natural sources also contribute to Arctic Haze during wintertime and the winter-spring transition *(Law et al., 2017; Schmale et al., 2018; Kirpes et al., 2019)*. During wintertime 14% of organic mass at Alert originated from gas flaring in northern Russia *(Leaitch et al., 2018)*. For example, gas flaring from Russia contributes to black carbon at Alert (northern Canada) and Utqiaġvik (northern Alaska) *(Stohl et al., 2013; Qi et al., 2017; Xu et al., 2017; Marelle et al., 2018)*. Metal industry and combustion sources, such as power generation, from Siberia (e.g. Kola peninsula) were identified as sources of pollution at Villum station, Greenland during winter and spring *(Nguyen et al., 2013)*. Metal smelting from Siberia also contributes to $SO_4^{2-}$ at Zeppelin during wintertime *(Hirdman et al., 2010)*. A more recent study by *Winiger et al. (2019)* showed that during wintertime Arctic sites, such as Utqiaġvik, Alert, Zeppelin, are influenced by fossil fuel combustion emissions. Petroleum extraction on the North Slope of Alaska, including Prudhoe Bay, was found to influence aerosol distributions, composition, and particle growth at Utqiaġvik, with enhanced growth of ultrafine particles *(Kolesar et al., 2017; Kirpes et al., 2018)*.

Natural aerosol sources also contribute to Arctic Haze such as dust, volcanic emissions and and sea-spray aerosols (SSA) *(Rahn et al., 1977; Barrie and Barrie, 1990; Quinn et al., 2002; Stone et al., 2014; Huang et al., 2015; Zwaaftink et al., 2016; Kirpes et al., 2018)*. Dust is not only transported from mid-latitudes sources (Asia, Africa), but it is also produced within the Arctic, with local dust contributing up to 85% to total dust burden in the Arctic *(Zwaaftink et al., 2016)*. During wintertime, fresh SSA (including sodium ions ($Na^+$), chloride ions ($Cl^-$), sea-salt (ss)-$SO_4^{2-}$ and marine organics) can be a significant fraction of particulate matter, 40% of super-micron (1 to 10 $\mu$m particle diameter) and 25% of sub-micron (up to 1$\mu$m particle diameter) *(Quinn et al., 2002)*. While studies have largely focused on anthropogenic sources of Arctic Haze influencing, in particular BC and $SO_4^{2-}$, there have been fewer studies on the contribution of SSA, the focus of this study. The primary mechanism leading to the formation of SSA is bubble bursting (jet-drop and film-drop formations) on the sea surface due to wind stress during whitecap formation *(Monahan et al., 1986)*. For this reason, wind speed is a significant parameter affecting SSA productivity *(Russell et al., 2010; Saliba et al., 2019)*. Arctic warming is leading to a decrease of sea-ice during summertime and, as a result, less and thinner sea-ice is forming during wintertime *(Stroeve et al., 2012)*. Thus, new SSA sources, such as open ocean and leads, may contribute more in the future to the total aerosol burden over Arctic coastal regions, impacting CCN concentrations and radiative forcing *(Ma et al., 2008)*.

A detailed analysis of in-situ aerosol composition in Utqiaġvik revealed that, due to long-range transport from the North Pacific (due to strong winds in source regions, such as in the Pacific Ocean), sub-micron SSA peaks in winter and early spring,



while super-micron SSA peaks in summer, due to sea-ice retreat *(Quinn et al., 2002)*. However in winter, super-micron SSA
mass concentrations increase in the presence of open leads, while sub-micron SSA appear to be more influenced by long-range
transport *(May et al., 2016; Kirpes et al., 2019)*. *Kirpes et al. (2018)* analysed atmospheric particle samples collected in winter
2014 in Utqiaġvik. They found that the samples were influenced by air masses from the Arctic Ocean to the north and Prudhoe
Bay oilfields to the east. Aged SSA were always internally mixed with secondary $SO_4^{2-}$, or with both $SO_4^{2-}$ and $NO_3^-$ and
reduced chlorine, suggesting anthropogenic influence from background Arctic Haze or Alaskan oil field emissions. *Kirpes
et al. (2019)* concluded that fresh SSA, based on the presence of $Na^+$ and $Cl^-$ in ratios similar to seawater, including marine
organic aerosols, were produced locally from open leads, with indications of secretions from sea ice algae and bacteria based
on observed enrichments in carbon/sodium ($C:Na^+$) ratios. Previous studies of the Arctic and North Atlantic during wintertime
and the winter-spring transition also showed that the majority of sub-micron organic mass (OM) is highly correlated with $Na^+$
concentrations *(Russell et al., 2010; Shaw et al., 2010; Frossard et al., 2011; Leaitch et al., 2018)*. Frost flowers with organic-
salt coatings have also been proposed as a possible source of wintertime SSA *(Xu et al., 2013)*, although *Kirpes et al. (2019)*
found no evidence of frost flowers or blowing snow as a potential source, supporting the findings of older studies *(Roscoe et al.,
2011)*.

Regional and global models have difficulties capturing wintertime Arctic Haze composition and often underestimate $SO_4^{2-}$
and BC *(Bond et al., 2013; Eckhardt et al., 2015; Sato et al., 2016; Schwarz et al., 2017; Whaley et al., 2022)*, while the
contribution of SSA to Arctic Haze remains poorly evaluated *(Kirpes et al., 2019)*. Representation of SSA concentrations in
models has been improved over recent years, but with less focus on the Arctic winter. For example, SSA source functions
with updated dependencies on wind speeds, sea surface temperatures (SSTs) or salinity *(Revell et al., 2019; Jaeglé et al., 2011;
Sofiev et al., 2011)* have led to improve simulation of super-micron SSA. However, sub-micron SSA is often still underestimated
*(Sofiev et al., 2011; Huang and Jaeglé, 2017)* and sub-micron emissions of SSA from frost flowers and blowing snow have
been included in models *(Xu et al., 2013, 2016; Huang and Jaeglé, 2017; Rhodes et al., 2017)*. Modelled SSA including a
source of frost flowers captures better monthly SSA concentrations at Alert during wintertime, while a source of blowing
snow overestimates observations *(Huang and Jaeglé, 2017; Marelle et al., 2021)*. At Utqiaġvik during January and February
a source of blowing snow improves modelled SSA; however it still cannot explain the high observed SSA, while the blowing
snow explains high observed SSA in the Antarctic *(Huang and Jaeglé, 2017)*.

In this study, the performance of the Weather Research Forecast model, coupled with chemistry (WRF-Chem), is examined
with regard to its ability to simulate Arctic Haze composition as well as SSA components, including ss-$SO_4^{2-}$ and marine
organics. The model is first evaluated against available data over the wider Arctic, and the sensitivity to more recent treatments
of SSA wind speed and SST dependencies, is investigated. Inclusion of a marine organic source is also examined *(Fuentes et al.,
2010, 2011)*. The findings of *Kirpes et al. (2019)* are used as a basis for a more focused regional study to evaluate modelled
Arctic wintertime aerosol composition in northern Alaska. The sensitivity of model results to processes influencing SSA
production and concentrations are investigated including aerosol dry deposition, wind speed dependence and sea-ice fraction.
Missing local sources of marine organics are also examined based on the findings of *Kirpes et al. (2019)*. A companion paper,





*Ioannidis et al., (2022)* (in prep.), examines the contribution of remote and regional anthropogenic emissions to Arctic BC in northern Alaska and northern Russia during wintertime.

The model setup, including the emissions are described in Section 2. The observed aerosol composition used to evaluate the model performance are introduced in Section 3. The model runs, including sensitivity simulations, together with results are presented in Sections 4 and 5. First, in Section 4, simulated Arctic Haze, focusing on SSA, is evaluated at remote Arctic sites. Second, in Section 5, the results from the regional study over northern Alaska during wintertime and sensitivity of results to processes influencing SSA production in the model are presented. The implications of our findings for the simulation of Arctic

Haze aerosols and conclusions are presented in Section 6.

## 2   WRF-Chem

### 2.1   Model Setup

WRF-Chem chemical transport model version 3.9.1.1 is used to simulate quasi-hemispheric and regional Arctic Haze aerosols and to examine local SSA sources over northern Alaska. WRF-Chem is a fully coupled, online meteorological and chemical

transport mesoscale model *(Grell et al., 2005; Fast et al., 2006)*. Recent improvements in the WRF-Chem model over the Arctic are included in the version used in this study *(Marelle et al., 2017)*. The model setup, including meteorological and chemical schemes, is shown in Table 1. Briefly, Yonsei University (YSU - boundary layer), Model Version 5 similarity (MM5 - surface layer) and Noah-Multiparameterization Land Surface Model (NOAH MP, land surface model) are used. More details about the NOAH MP scheme are given in APPENDIX A.

All the various processes for aerosols in the atmosphere, like nucleation, evaporation, coagulation, condensation, dry deposition, aerosol/cloud interactions and aqueous chemistry, are included in the Model for Simulating Aerosol Interactions and Chemistry (MOSAIC, *Zaveri et al. (2008)*) scheme. MOSAIC treats all the major aerosol species, such as $SO_4^{2-}$, $NO_3^-$, $Cl^-$, ammonium ($NH_4^+$), $Na^+$, BC, and OA. The size distribution of each aerosol species is represented by eight bins, from 39 nm to 10 $\mu$m. Each bin is assumed to be internally mixed, and both mass and number are simulated. The applied MOSAIC version

includes secondary organic aerosol formation (SOA) from the oxidation of anthropogenic and biogenic species *(Shrivastava et al., 2011; Marelle et al., 2017)* and is combined with SAPRC-99 gas-phase chemistry. In the base model, OA is the sum of SOA and anthropogenic emissions of organic matter (OM). Aerosol sedimentation in MOSAIC is calculated throughout the atmospheric column based on the Stokes velocity scheme, as described in *Marelle et al. (2017)*.

### 2.2   Emissions

This section provides details about the emissions that are used in the simulations. More details are provided about SSA emissions since this is the focus of this study.



**Table 1.** WRF-Chem model setup.

| Parametrization scheme | Options |
| --- | --- |
| | Physics (WRF) |
| Planetary boundary layer | Yonsei University (YSU) - *(Hong et al., 2006)* |
| Surface layer | Pennsylvania State / NCAR Mesoscale |
| | Model Version 5 (MM5) similarity *(Grell et al., 1994; Jiménez et al., 2012)* |
| Land surface | NOAH MP *(Niu et al., 2011)* |
| Microphysics | Morrison *(Morrison, 2009)* |
| SW & LW radiation | Rapid Radiative Transfer Model (RRTMG - *Iacono and D. (2008)*) |
| Cumulus parameterization | Kain-Fritsch with cumulus potential (KF-CuP) |
| | *(Berg et al., 2013)* |
| | Chemistry (WRF-Chem) |
| Aerosol model | MOSAIC 8-bins *(Zaveri et al., 2008)* |
| Gas-phase chemistry | Statewide Air Pollution Research Center SAPRC-99 |
| | modified with added dimethyl sulphide chemistry *(Carter, 2000; Marelle et al., 2017)* |
| Photolysis | Fast-J *(Wild et al., 2000)* |
| Sea-spray emissions | *Gong et al. (1997)* |

### 2.2.1 Anthropogenic and natural emissions

Anthropogenic emissions are from the Evaluating the Climate and Air Quality Impacts of Short-Lived Pollutants version 6 (ECLIPSE v6b) inventory, with a resolution of 0.5º x 0.5º *(Whaley et al., 2022)*, including emissions of OM. Emissions of

dimethyl sulfide (DMS), mineral dust, and lightning $NO_x$ are calculated online in the model (see *Marelle et al. (2017)* and references therein). Biogenic emissions for 2014 are calculated online using Model of Emissions of Gases and Aerosol from Nature (MEGAN) model *(Guenther et al., 2012)*.

### 2.2.2 Sea-spray emissions

In the control simulation, sea-salt emissions of $Na^+$ and $Cl^-$ are included. They are calculated per particle radius, with 1000

sub-bins per MOSAIC bin, using the density function dF/dr (in particles m$^{-2}$ s$^{-1}$ $\mu$ m$^{-1}$) from *(Gong et al., 1997)* (G97 from now on) which represents the rate of seawater droplets form per unit area (sea surface) and per increase of particle radius and its derived from the source function based on laboratory experiments described in *Monahan et al. (1986)* (MO86 from now on):

$$\frac{dF}{dr} = 1.373 \times U_{10}^{3.41} \times r^{-3}(1 + 0.057 \times r^{1.05}) \times 10^{1.19e^{-B}} \tag{1}$$

where F is a function of U and r, r is the particle radius at relative humidity (RH) equal to 80%, U the 10m-elevation wind speed and $B = \frac{(0.380 - logr)}{0.650}$. The source function is applied for particles with dry diameters of 0.45 $\mu$m or more. For particles





with dry diameters less than 0.45 $\mu$m, a correction is applied to the formula based on *O'Dowd et al. (1997)*. This approach is based on the whitecap method, where the emission flux scales linearly with the fraction of ocean area covered by whitecaps. Over open ocean, the whitecap fraction, W(U), is determined as a function of wind speed (*Monahan and Muircheartaigh*

*(1980)*; MO80 from now on):

$$W(U) = 3.84 \times 10^{-6} \times U_{10}^{3.41} \tag{2}$$

This expression for W(U) is included implicitly in Equation (1) following details provided in MO80. In the base version of WRF-Chem SSA emissions are calculated for every grid cell, which is open ocean or salt-water lakes. In this study, the grid cell which is covered by sea-ice is considered and then the fraction of that ice-free grid is used. In this way, SSA emissions

from open leads are taken into account.

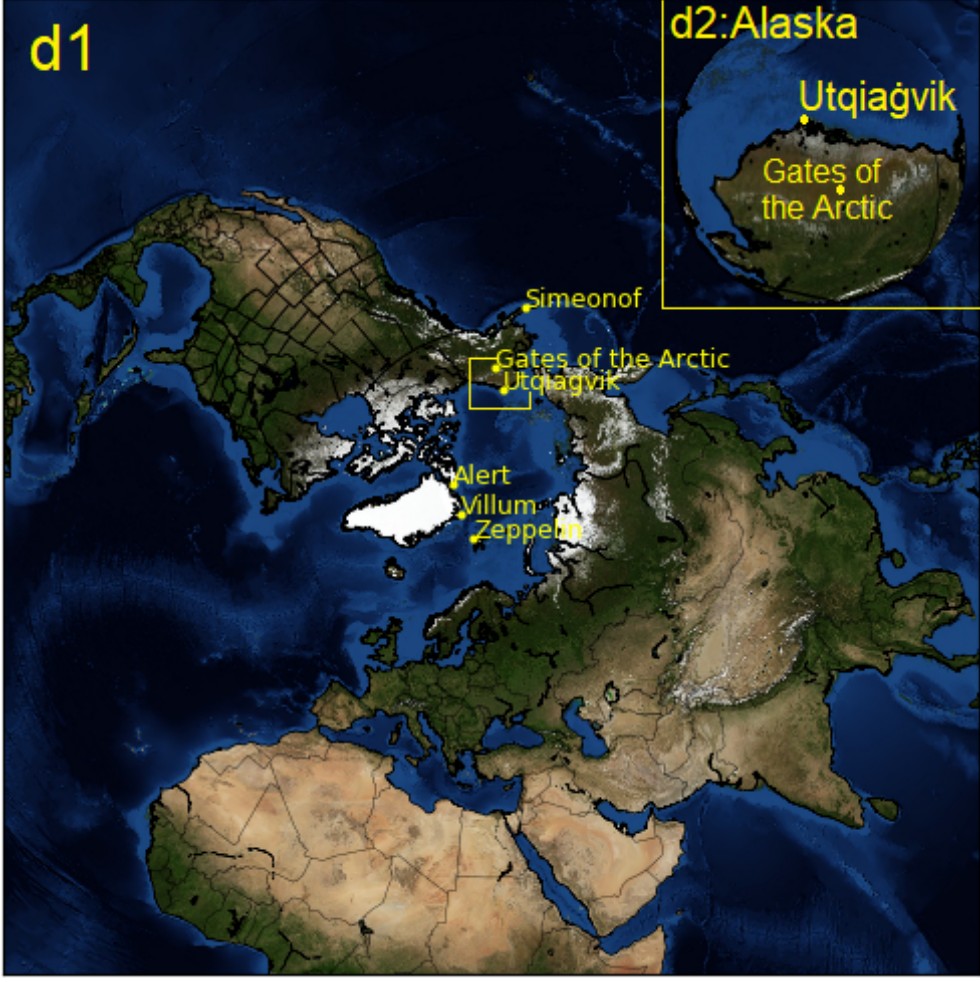

**Figure 1.** WRF-Chem simulation domains: d1 is the 100km domain and d2 is the 20km domain.



## 2.3    Simulations

Two simulation domains on a polar stereo-graphic projection are used in this study, as shown in Figure 1. The first (parent) domain (d1) covers a large part of the Northern Hemisphere with $100 \times 100$ km horizontal resolution. The boundary and initial conditions, are derived from National Centres for Environmental Prediction Final meteorological reanalysis data (NCEP FNL

$1^{\circ}x1^{\circ}$), (*National Centers for Environmental Prediction, National Weather Service, NOAA, U.S. Department of Commerce (2000)* and Model for OZone And Related chemical Tracers (MOZART, *Emmons et al. (2010)*) for atmospheric trace gases and aerosols. The nested domain (d2), run at horizontal resolution of $20 \times 20$ km, covers continental Alaska, a small area of northwest Canada, and the Chukchi and Beaufort Seas (see Figure 1). 50 vertical levels and grid nudging are used for the 100 km resolution domain, while calculating spectral nudging parameters as in *Hodnebrog et al. (2019)*, is implemented in the

nested domain. WRF-Chem temperatures and winds are nudged at each dynamical step to the reanalysis, which are updated every 6 hours, above the atmospheric boundary layer.

The simulations performed in this study are discussed in sections 4 and 5. Simulations at 100 km are run for 4 months from November 2013 until the end of February 2014, with the first two months considered as spin-up. The model is run at 20 km for two different periods (23–28 January 2014 and 24–28 February 2014) corresponding to the campaign which took place in

Utqiaġvik, and described earlier (*Kirpes et al. (2018, 2019)*, KRP18 and KRP19 from now on, respectively, see also sub-section 3.2). For these simulations, the initial, and boundary conditions are derived from the quasi-hemispheric simulation. A series of sensitivity runs are performed to examine processes affecting SSA emissions over northern Alaska. They are summarized in Table 3 and discussed in detail in Section 5. At 20 km for all the simulations, 4 days prior to the beginning of the campaign considered spin up. In all runs, the model results are output every 3h.

## 3    Aerosols

### 3.1    Routine monitoring sites

Surface mass concentration data (for aerodynamic diameters (defined as $d_a$) $\leq 2.5$ $\mu$m and $d_a < 10$ $\mu$m), from EMEP (European Monitoring and Evaluation Programme) dataBASe (EBAS - http://ebas.nilu.no) for Zeppelin, Ny-Ålesund, Norway (78.9N, 11.88W) and Alert, Canada (82.5N, -63.3W), are used to evaluate the quasi-hemispheric model simulations together with

data from Villum Research Station, Station Nord, Greenland (81.6N, -16.7W), referred to as Villum from now on (reporting total suspended particulates). The data are collected on a daily (Zeppelin) and weekly (Villum, Alert) basis. At Alert, and Zeppelin *(Aas et al., 2021)*, observations for $Na^+$, $Cl^-$, $NH_4^+$, $NO_3^-$ and $SO_4^{2-}$ measured with ion chromatography are used *(Sharma et al., 2019)*. At Villum, the same observations are collected using a filter-pack over a week and analysed using by ion-chromatography ($Cl^-$, $SO_4^{2-}$), cat-ion ionchromatograph ($Na^+$) and segmented flow analysis ($NH_4^+$). For all the EBAS

stations, the units of inorganic aerosols ($NH_4^+$, $SO_4^{2-}$, $NO_3^-$) are converted to model units ($\mu gm^{-3}$), using the ratio of molar weights of $NH_4^+$, $NO_3^-$, $SO_4^{2-}$ to molar weights of nitrogen or sulphur, respectively. With regard to measurement uncertainties,





EBAS documentation notes, in the case of Alert only, that there are uncertainties of around 33% and 36% in Na$^+$, SO$_4^{2-}$, NO$_3^-$ and Cl$^-$, respectively, and higher uncertainties (43%) for NH$_4^+$.

Surface mass concentration data, diameter less than 2.5 $\mu$m (r$_d$ ≤ 2.5 $\mu$m), from the Interagency Monitoring for Protected Visual Environments (IMPROVE) database is also used for model evaluation for Simeonof (55.3N, -160.5W), a sub-Arctic site on the Aleutians islands, south of Alaska and an inland site, Gates of the Arctic (66.9N, -151.5W) which is located south-east of Utqiaġvik. The samples are collected on-site (e.g. Simeonof site) over 24 hours every three days (http://views.cira.colostate.edu/fed/QueryWizard/Default.aspx, *Malm et al. (1994)*). At these two sites observations of Na$^+$, Cl$^-$, OC, NO$_3^-$ and SO$_4^{2-}$ are used. To compare with the OC observations at the two Alaskan sites modelled OA is divided by 1.8, the reported ratio of OM/OC in the documentation for these two stations *(Malm et al., 1994)*. In this study, mass concentration data with diameter ≤ 2.5 $\mu$m are defined as fine mode aerosols, while diameter < 10 $\mu$m then are defined as coarse mode aerosols.

Sub-micron (d$_a$ < 1.0$\mu$m) and super-micron (1.0 < d$_a$ < 10 $\mu$m) surface mass concentration data from the National Oceanic and Atmospheric Administration (NOAA) Barrow observatory (71.3N, -156.8W), near Utqiaġvik town, is also used in this study, with daily and weekly temporal coverage, respectively. The sampling site is located 8 km northeast of Utqiaġvik, 20 m above mean sea level (msl), with a prevailing, east-northeast wind off the Beaufort Sea. Concentration data (Na$^+$, Cl$^-$, NH$_4^+$, NO$_3^-$ and SO$_4^{2-}$) are determined by ion chromatography *(Quinn et al., 1998)* and are sampled only for wind directions between 0 and 130 degrees (with 0 degrees indicating north). According to *Quinn et al. (2002)* measurement uncertainties of SSA components and SO$_4^{2-}$ are below 1%, while for NH$_4^+$ they are 7.8%. Observed ss-SO$_4^{2-}$ is calculated from observed Na$^+$ concentrations and the mass ratio of SO$_4^{2-}$ to Na$^+$ in seawater of 0.252 *(Bowen et al., 1979; Calhoun et al., 1991)*.

The model Stokes diameter (r$_d$) is converted to aerodynamic diameter using the *Seinfeld and Pandis (1998)* formula. Thus, the diameter of modelled sub-micron particles is up to 0.73 $\mu$m (including the first four MOSAIC bins and a fraction of the 5th bin), and super-micron particle diameters are between 0.73 to 7.3 $\mu$m (fraction 5th bin, 6th and 7th bins and fraction 8th bin). Seven MOSAIC bins and a fraction of the 8th bin are used (modelled stokes r$_d$ ≤ 7.3 $\mu$m) to compare with Alert and Zeppelin observations (aerodynamic d$_a$ < 10 $\mu$m, coarse mode). All model aerosol bins are used to compare with observations at Villum, where the observations are reported as total suspended particulates (TSP), i.e. there is no cutoff. For each site, modelled aerosols are estimated at the same conditions (temperature, pressure) as the reported observations. Overall, particles at different size ranges (up to 1.0 $\mu$m, 2.5 $\mu$m, and 10 $\mu$m) are used to validate the model performance in each domain.

### 3.2 Campaign data

Details about the field campaign (January 23–27 and February 24–28, 2014) measurements near Utqiaġvik, Alaska can be found in KRP18 and KRP19. Briefly, atmospheric particles were collected using a rotating micro-orifice uniform deposition impactor located 2 m above the snow surface at a site located 5 km across the tundra from the NOAA Barrow Observatory and inland from the Arctic Ocean. The sampled particles were analysed by computer-controlled scanning electron microscopy with energy scattering X-ray spectroscopy (CCSEM-EDX) to determine the individual particle morphology and elemental composition. The analysed samples were collected either during daytime or nighttime and only when wind directions were



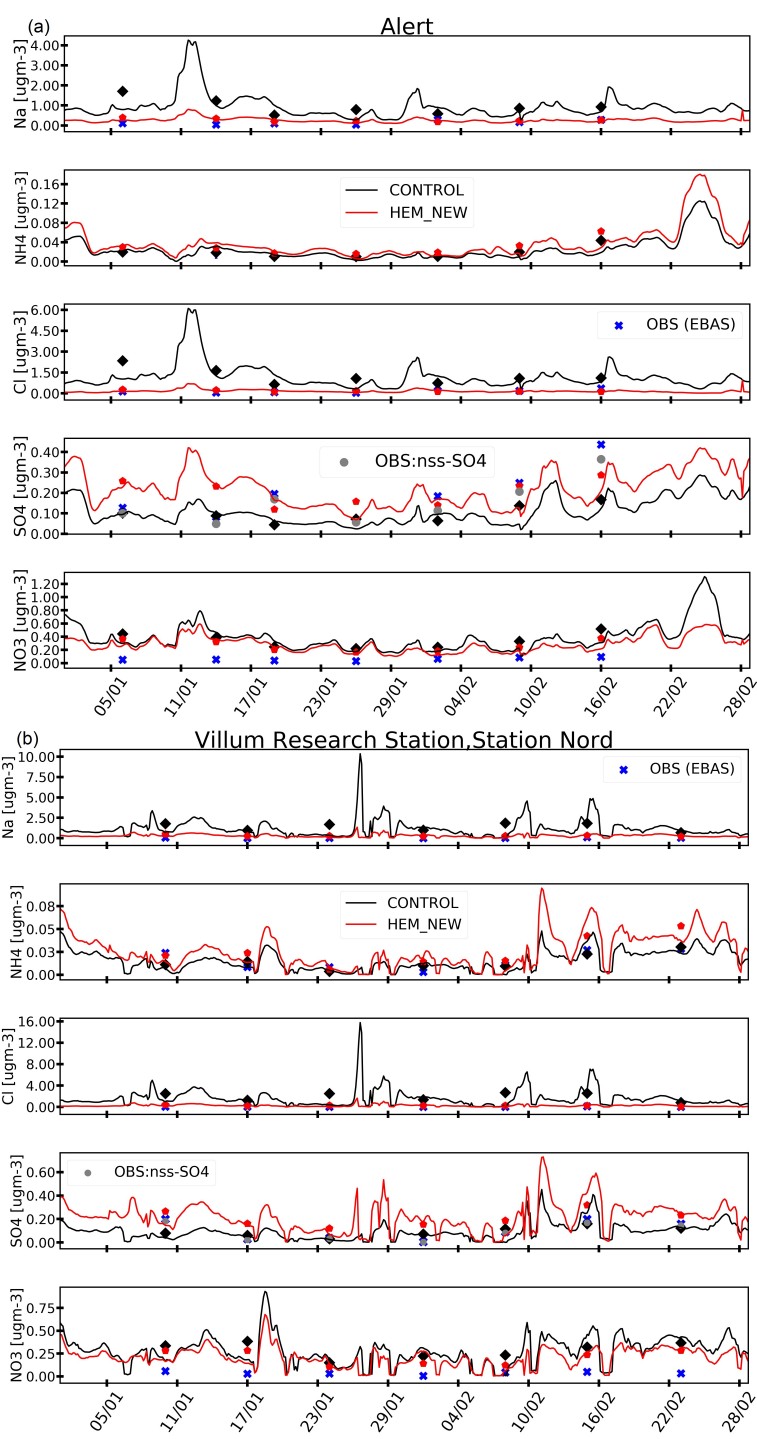



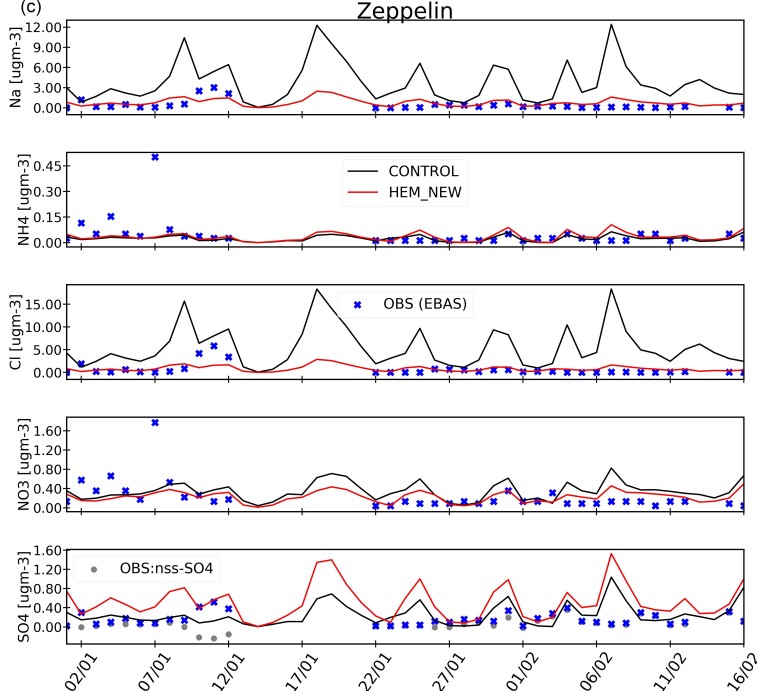

**Figure 2.** Evaluation of modelled aerosol composition (runs CONTROL and HEM_NEW) against in-situ observations of (a) coarse mode aerosols ($d_a < 10\,\mu$m) at Alert, Canada (standard temperature pressure (STP) conditions), (b) TSP aerosols ($d_a \leq 10\,\mu$m) at Villum, Greenland and (c) coarse mode aerosols ($d_a < 10\,\mu$m) at Zeppelin, Norway in UTC. The black line shows model results from the CONTROL run; the red line shows the HEM_NEW run, while observations are shown as blue crosses. Villum and Alert observations are weekly averages, and the corresponding model weekly averages are shown as black diamonds for CONTROL and red pentagons for HEM_NEW. Zeppelin observations are daily 24h averages. Observations are shown only when they are available. See the text for details about the observations and model runs.

between 75 and 225 degrees, corresponding to minimise local pollution influence. Data analysis provided information about the different chemical components as a fraction of the total number of particles sampled.

## 4   Processes influencing SSA over the wider Arctic and their contribution to wintertime Arctic aerosols

This section focuses on evaluating the capability of the model to simulate Arctic Haze aerosols during wintertime and improv-
ing model treatments of SSA. Briefly, in the base simulation (CONTROL), sea-salt emissions are calculated using the G97

parametrization scheme, including the MO80 whitecap method. All the updates described below are included in a new quasi-
hemispheric simulation (HEM_NEW) with the aim to improve the model. This includes addition of marine organics *(Fuentes
et al., 2010)*, using a more recent whitecap method (Salisbury et al., 2014), including the dependence of SSA emissions on
SST (Jaeglé et al., 2011), and the addition of a ss-SO$_4^{2-}$ component, based on Kelly et al. (2010). HEM_NEW simulation is



then evaluated (sub-section 4.6) compared to CONTROL and the observations at the different sites followed by a discussion of
the new results.

## 4.1  Anthropogenic and natural aerosols in the Arctic

First considering the observations, at remote sites such as Alert (Fig 2a), observed Na$^+$, Cl$^-$ and NO$_3^-$ coarse mode mass
concentrations do not exceed 0.3, 0.5 and 0.09 $\mu$ g m$^{-3}$, respectively, during the study period. Total SO$_4^{2-}$ (sum of ss-SO$_4^{2-}$ and
nss-SO$_4^{2-}$) reach 0.44 $\mu$ g m$^{-3}$, which is mostly nss-SO$_4^{2-}$, as ss-SO$_4^{2-}$ does not exceed 0.09 $\mu$gm$^{-3}$, likely to be due to long-rage
transport from sources in north-central, western, northwest Russia and Europe *(Leaitch et al., 2018)*. Similar magnitudes have
been reported in previous studies during winter months *(Leaitch et al., 2018)*. NH$_4^+$ peaks at 0.06 $\mu$gm$^{-3}$ and originates from
Russia and Europe during winter *(Leaitch et al., 2018)*. At Villum (Fig 2b), observed TSP Na$^+$, Cl$^-$ and NO$_3^-$ reach up to 0.12,
0.13 and 0.06 $\mu$ g m$^{-3}$, respectively. These concentrations are lower than at Alert which could be explained by the fact that
during winter the sea surrounding Villum station is frozen *(Nguyen et al., 2013)*. Total SO$_4^{2-}$ does not exceed 0.2 $\mu$ g m$^{-3}$ and is
mostly nss-SO$_4^{2-}$ (up to 0.18 $\mu$ g m$^{-3}$), while ss-SO$_4^{2-}$ does not exceed 0.03 $\mu$ g m$^{-3}$. At Villum, SO$_4^{2-}$ peaks during wintertime
*(Massling et al., 2015)* and is the dominant component of Arctic Haze at this site *(Lange et al., 2018)*. NH$_4^+$ concentrations at
Villum are up to 0.1 $\mu$ g m$^{-3}$. In the high Arctic, Na$^+$ could potentially also originate from anthropogenic sources which could
account for up to 35% of total Na$^+$ *(Barrie and Barrie, 1990)*. Note that this source is not included in the model, or in models
generally. Higher Na$^+$, Cl$^-$ and NO$_3^-$ concentrations are observed at Zeppelin (coarse mode) reaching up to 3.0, 5.9 and 1.8
$\mu$ g m$^{-3}$, respectively (Fig. 2c). Total SO$_4^{2-}$ does not exceed 0.8 $\mu$ g m$^{-3}$ and might originate from metal smelting in Siberia
*(Hirdman et al., 2010)*. ss-SO$_4^{2-}$ contributes up to 0.8 $\mu$ g m$^{-3}$ of the total SO$_4^{2-}$. Note that, in some cases, nss-SO$_4^{2-}$ has small
negative concentrations, due to depletion of ss-SO$_4^{2-}$ through fractionation processes *(Quinn et al., 2002)*. Observed NH$_4^+$ does
not exceed 0.5 $\mu$ g m$^{-3}$ during the study period.

At Simeonof, an ice-free sub-Arctic island in south western Alaska, high concentrations of fine mode Na$^+$ and Cl$^-$ are
observed of up to 2.1 and 1.0 $\mu$ g m$^{-3}$, respectively (Fig. 3a), especially at the beginning of January 2014, with low values of
NO$_3^-$ (peaking at 0.25 $\mu$ g m$^{-3}$). Total SO$_4^{2-}$ reaches 1.0 $\mu$ g m$^{-3}$ and is mostly nss-SO$_4^{2-}$ (0.9 $\mu$ g m$^{-3}$), while the contribution
of ss-SO$_4^{2-}$ is smaller (up to 0.3 $\mu$ g m$^{-3}$). Lower concentrations of fine mode Na$^+$ and Cl$^-$ (up to 0.35 $\mu$ g m$^{-3}$) are observed at
Gates of the Arctic (Fig. 3b), a non-coastal site located 404 km south-east of Utqiaġvik in the Brooks Range Mountains, while
NO$_3^-$ peaks at 0.45 $\mu$ g m$^{-3}$. Total SO$_4^{2-}$ peaks at 0.64 $\mu$ g m$^{-3}$ and 0.56 $\mu$gm$^{-3}$ is nss-SO$_4^{2-}$ possibly due to local anthropogenic
emissions originating from the North Slope of Alaska oilfields which may affect the measurements although this site is located
inland (391 km) south of the oilfields. The contribution of ss-SO$_4^{2-}$ is insignificant (no more than 0.08 $\mu$ g m$^{-3}$) at this site.

At Utqiaġvik, observed super-micron (1.0 < d$_a$ < 10.0 $\mu$m) Na$^+$ and Cl$^-$ concentrations reach 1.2 $\mu$ g m$^{-3}$ (Fig. 4b), while
NO$_3^-$ peaks at 0.2 $\mu$ g m$^{-3}$. Super-micron SO$_4^{2-}$ and NH$_4^+$ do not exceed 0.16 and 0.009 $\mu$ g m$^{-3}$, respectively. Super-micron
NH$_4^+$ concentrations are insignificant *(Quinn et al., 2002)*. However, there is more ss-SO$_4^{2-}$ (up to 0.18 $\mu$gm$^{-3}$) than nss-
SO$_4^{2-}$. On the other hand, observed sub-micron Na$^+$, Cl$^-$ and NO$_3^-$ at Utqiaġvik peak at 2.0, 2.2, and 0.9 $\mu$ g m$^{-3}$ respectively
(Fig. 4a). Note that based on the findings of KRP18, only 1%, by number, of the particles across the 0.15-1.0 $\mu$m size range
corresponded to fly ash and dust, as compared to 50-90% from SSA across the same size range. This supports the assumption





of Na$^+$ being primarily from SSA during this study. High sub-micron observed total SO$_4^{2-}$ (mostly nss-SO$_4^{2-}$) concentrations were measured at Utqiaġvik and peak at 2.4 $\mu$gm$^{-3}$, possibly due to local influence from Prudhoe Bay oil fields to the east

(KRP18, KRP19), a magnitude much higher than super-micron SO$_4^{2-}$, also reported for Utqiaġvik by *Quinn et al. (2002)*. Enhanced nss-SO$_4^{2-}$ during this period at Utqiaġvik could also be due to transport from mid-latitude sources, as well as due to transport and oxidation of SO$_2$ to SO$_4^{2-}$ near and within the Arctic region *(Barrie and Hoff, 1984)*. Sub-micron ss-SO$_4^{2-}$ peaks at 0.5 $\mu$ g m$^{-3}$. Observed NH$_4^+$ is higher compared to the other remote Arctic sites (up to 0.34 $\mu$ g m$^{-3}$). NH$_4^+$ temporal variation during January and February follows that of nss-SO$_4^{2-}$ due to NH$_3$ reaction with acidic SO$_4^{2-}$ aerosol near source

regions outside of the Arctic *(Quinn et al., 2002)* or due to regional sources of NH$_3$, e.g. combustion of fossil fuels *(Whaley et al., 2018)*.

Finally, only two sites provide total organic carbon (tOC) observations. Here, observed total organic carbon is assumed to include secondary organic aerosols, anthropogenic organic carbon emissions and marine organics. Thus, from now on it will be referred as tOC, to distinguish from OA and OM defined earlier. tOC ranges between 0.15 and 0.3 $\mu$ g m$^{-3}$ at Simeonof and

0.15 and 0.5 $\mu$gm$^{-3}$ at Gates of the Arctic during January and February 2014.

Evaluation of the CONTROL simulation shows that the model overestimates observed fine/coarse mode, super-micron and TSP Na$^+$ and Cl$^-$ at most sites, and especially at Simeonof (by up to 15 $\mu$ g m$^{-3}$), Zeppelin (Fig. 2c) (by up to 5.0 $\mu$ g m$^{-3}$), Utqiaġvik (by up to 0.3 $\mu$ g m$^{-3}$) and Gates of the Arctic (Fig. 3b) (by up to 4.0 $\mu$ g m$^{-3}$) site. The CONTROL simulation also overestimates NO$_3^-$ by up to 0.5 $\mu$gm$^{-3}$ at each site. On the other hand, this simulation captures NH$_4^+$ variability quite well

at Alert, Villum and Utqiaġvik (super-micron) (see also biases and RMSEs (Root Mean Square Error) in APPENDIX C and Tables C1, C2 and C7 respectively), whilst it overestimates NH$_4^+$ at Zeppelin by up to 0.4 $\mu$ g m$^{-3}$. CONTROL includes only the nss-SO$_4^{2-}$ component, however it captures observed variability of total SO$_4^{2-}$ at Zeppelin (coarse mode), Villum (TSP) and Utqiaġvik (super-micron), but underestimates total SO$_4^{2-}$ at Gates of the Arctic (fine mode) and Alert (coarse mode) by 0.5 and 0.2 $\mu$ g m$^{-3}$, respectively. In addition, the model underestimates sub-micron Na$^+$, Cl$^-$, SO$_4^{2-}$ and NH$_4^+$ at Utqiaġvik.

It also underestimates OA at the two sites compared to the measurements. In the following sections, model improvements are described. Biases and RMSEs in $\mu$gm$^{-3}$, are given in APPENDIX C for all sites and available aerosol species at each location.

## 4.2 Marine organics

Recent data-analysis studies *(Saliba et al., 2019; Kirpes et al., 2019)*, have suggested that marine organics contribute significantly to natural aerosol composition as ocean biomass can influence SSA number concentrations and diameter. In the

CONTROL run, marine organics are not activated; however a source code is included in the model by *Archer-Nicholls et al. (2014)*. For this reason, the parameterization, based on *Fuentes et al. (2010, 2011)* (F10 and F11 from now on, respectively) is activated in the MOSAIC scheme to include a source flux for marine organics with dry diameters up to 0.45 $\mu$m. The scheme is based on an analysis of data from a cruise in mid-latitudes investigating the influence of dissolved organic matter on the production of sub-micron SSA. The F10 SSA source function also depends on MO80 whitecap coverage. In this

study, organic fractions equal to 0.2 for the first and second MOSAIC bins, 0.1 for the third bin and 0.01 for the remaining bins are used following the high biogenic activity scenario which assumes high C:Chlorophyll-a (Chl-a) ratios. F11 found



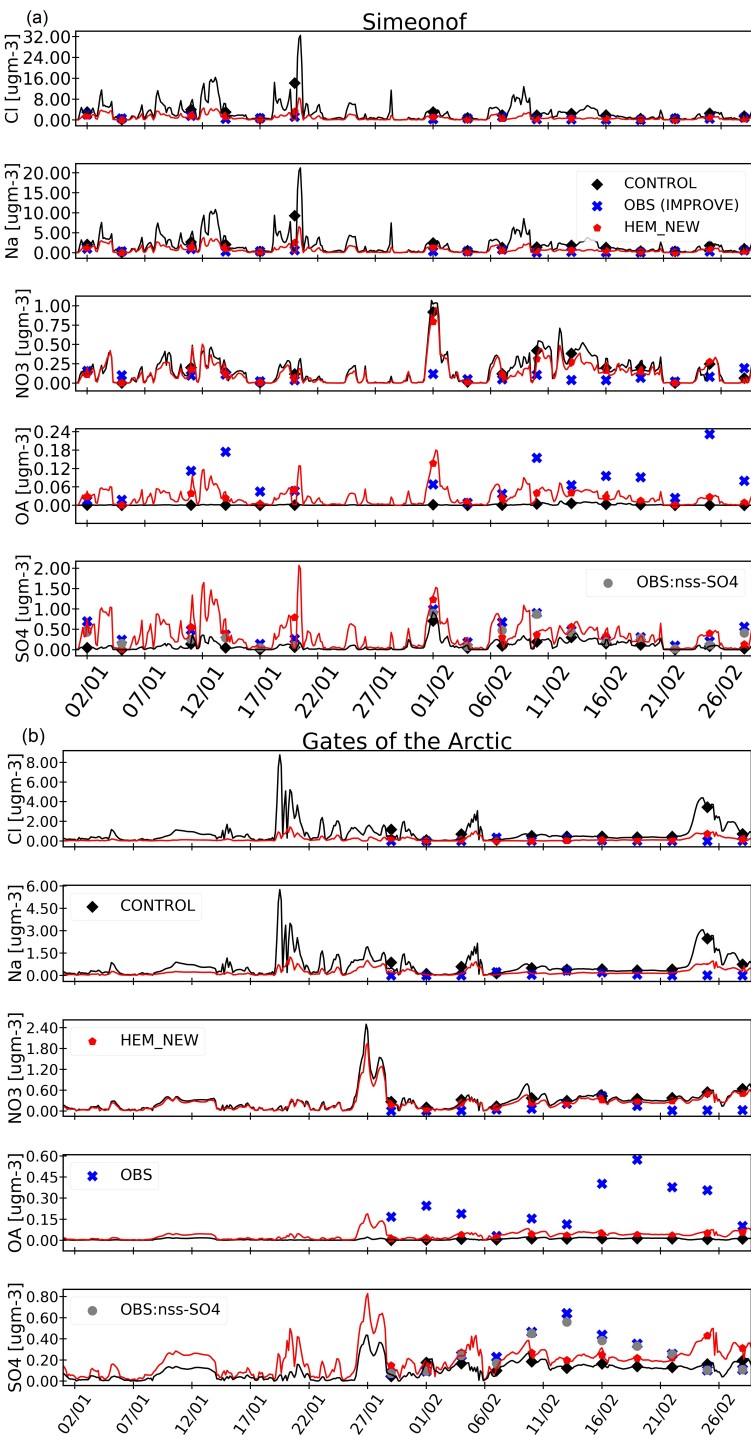

**Figure 3.** Evaluation of modelled aerosol composition (runs CONTROL and HEM_NEW) against in-situ aerosol observations of fine mode ($r_d \leq 2.5\ \mu$m) (both sites) at (a) Simeonof, Aleutians Islands, Alaska and (b) Gates of the Arctic, north of Alaska in local Alaskan time (AKST). The black line shows model results from the CONTROL run; the red line shows the HEM_NEW run, while observations are shown as blue crosses. Simeonof and Gates of the Arctic observations are 24h averages every three days and the corresponding model daily averages are shown as black diamonds for CONTROL and red pentagons for HEM_NEW. Observations are shown only when they are available. See the text for details about the observations and model runs.





that higher particle organic fractions are expected in algal bloom regions with high C:Chl-a ratios and Chl-a varying between 0.4-10 $\mu$g/L. The use of the F11 high biogenic activity option in our simulations is justified since MODIS-Aqua satellite data (https://neo.sci.gsfc.nasa.gov/view.php?datasetId=MY1DMW_CHLORA&date=2014-12-01) for January-February 2014

show that Chl-a south of Alaska and along the west coast of the United States varied between 0.3 and 3.0 $\mu$g/L. *Fujiki et al. (2009)* also found that Chl-a varied between 0.4 and 1.0 $\mu$g/L at six stations south of the Aleutian Islands, Alaska, during a sub-Arctic cruise in autumn 2005. Details about the F10 SSA source function are given in APPENDIX B. In the case of the model uses a source for marine organics, then OA is the sum of SOA, anthropogenic emissions of OM and marine organics. A more detailed analysis of marine organics, focusing on northern Alaska, is presented in 5.2.

## 295    4.3   Whitecap method

In agreement with previous modelling studies, e.g. *(Jaeglé et al., 2011)*, JA11 from now on and *(Spada et al., 2013; Revell et al., 2019; Hartery et al., 2020)* the CONTROL simulation produces too much coarse mode and TSP Na$^+$ and Cl$^-$. The G97 parametrization, which depends on the whitecap method and thus has a high wind speed dependence (see Eq. 1), has been widely adopted to simulate SSA emissions in global and regional models, e.g. JA11 and *Barthel et al. (2019)*. Several

studies tried to improve upon the whitecap method (W($U_{10}$)), especially for super-micron SSA. *Callaghan et al. (2008)* used an automated whitecap extraction technique to derive two whitecap expressions that differ from MO80, which are based on cubed relationships for $U_{10}$). Other factors, such as the wave field *(Salisbury et al., 2013)*, surfactant (amphiphilic organic material) activity *(Callaghan, 2013)* and fetch-dependent threshold for breaking waves *(Revell et al., 2019; Hartery et al., 2020)*, have also been shown to affect whitecap lifetime, with implications for SSA production. *Goddijn-Murphy et al. (2011)*

analysed Marine Aerosol Production (MAP) whitecap data, in combination with analysis of in-situ and satellite data (from Quick Scatterometer, QuikSCAT) for winds and waves. The satellite data were used to derive an expression with a lower wind speed dependence compared to MO80 (*Salisbury et al. (2014)*, SALI14 from now on). Here, the SALI14 parametrization is implemented instead of the MO80 whitecap fraction expression:

$$W(U) = 4.60 \times 10^{-3} \times U_{10}^{2.26} \tag{3}$$

Based on Figure 2 in SALI14, the seasonal mean of W($U_{10}$) using Eq. 3 is lower at high latitudes compared to MO80 during autumn and winter. By using this more recent whitecap fraction expression in the quasi-hemispheric simulation, super-micron SSA concentrations decrease overall within the Arctic (not shown here). More specifically, super-micron Cl$^-$ and Na$^+$ decrease more south of Alaska, by up to 20 $\mu$ g m$^{-3}$ (Aleutians Islands) and less north of Alaska, by up to 0.5 $\mu$ g m$^{-3}$. NO$_3^-$ also decreases slightly over continental Alaska, by up to 0.5 $\mu$ g m$^{-3}$, due to increased heterogeneous formation on SSA.

## 315    4.4   SST dependence

Recent data-analysis studies *(Saliba et al., 2019; Liu et al., 2021)* pointed out that wind speed alone cannot predict SSA variability, and it is important also to include a dependence on SSTs for SSA prediction. Recent modelling studies *(Jaeglé et al., 2011; Sofiev et al., 2011; Spada et al., 2013; Barthel et al., 2019)* showed that the application of SST dependence improves



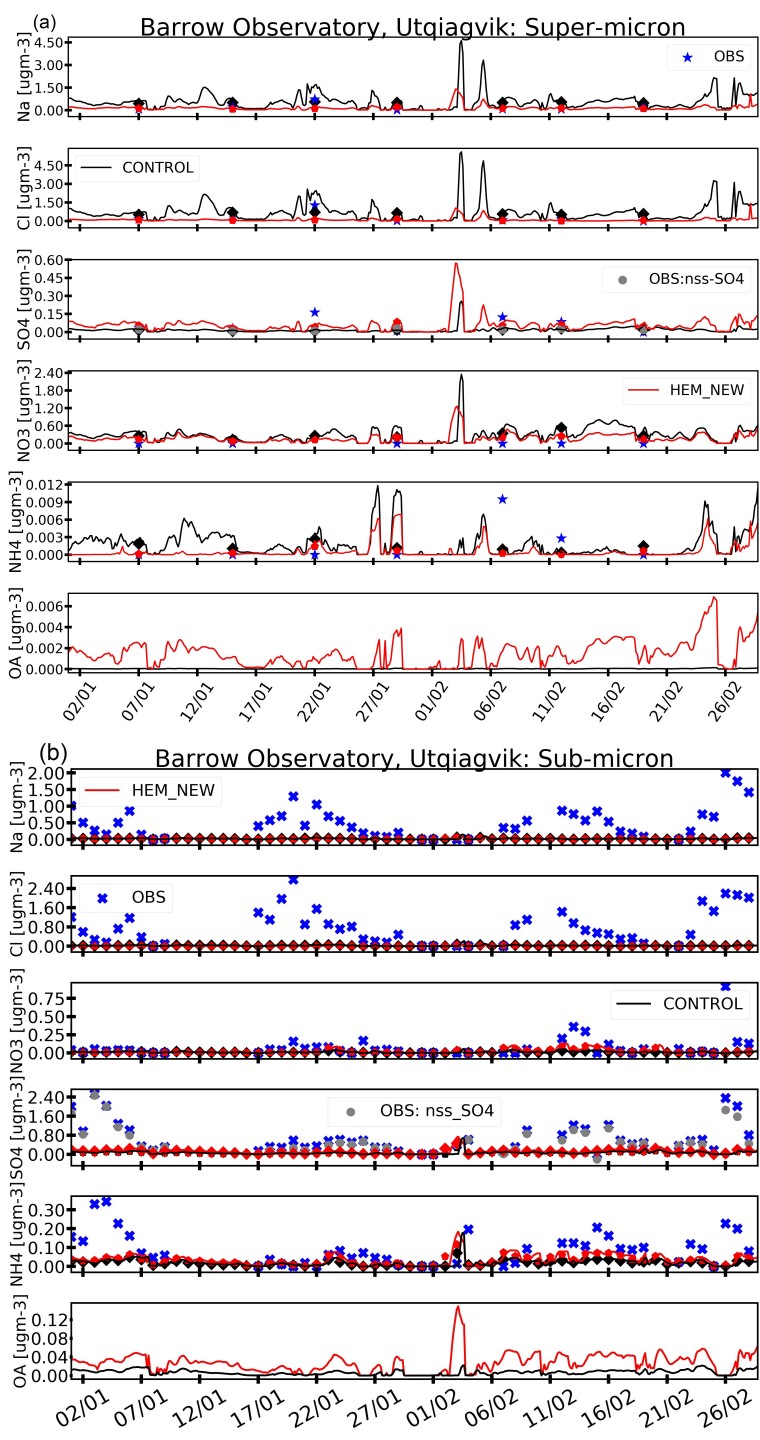

**Figure 4.** Evaluation of modelled aerosol composition (runs CONTROL and HEM_NEW) against in-situ observations at Barrow Observatory, near Utqiaġvik, Alaska for (a) super-micron and (b) sub-micron in UTC and in STP conditions. The black line shows model results from the CONTROL run; the red line shows the HEM_NEW run, while observations are shown as blue crosses. Sub-micron observations are daily averaged and super-micron observations are weekly averages. The corresponding model daily/weekly averages are shown as black diamonds for the CONTROL simulation and as red pentagons for the HEM_NEW. Observations are shown only when there are available. See the text for details about the observations and model runs.





simulated SSA concentrations compared to observations. More specifically, previous studies *(Spada et al., 2013; Grythe et al.,*
*2014; Barthel et al., 2019)* tested different SSA source functions, with and without SST dependence, and reported that including
such a dependence improves model results, regardless of the SSA source function employed. However, uncertainties still remain
about the role of SSTs on SSA production *(Revell et al., 2019)*, including the role of other factors, such as seawater composition
*(Callaghan et al., 2014)* or wave characteristics (e.g. wave speed and breaking wave type, *Callaghan et al. (2012))*, which might
be more important than SSTs alone. In this study, the JA11 SST correction factor is applied when SSTs are between -2°C and
30°C to evaluate the effect of SST on sub- and super-micron SSA emissions in the Arctic. In our simulations, SSTs are provided
by reanalyses data, in this case, FNL, and in the presence of sea-ice, SST is set equal to -1.75°C.

### 4.5 Sea-salt sulphate

Standard versions of the WRF-Chem model do not include ss-$SO_4^{2-}$. The Community Multiscale Air Quality (CMAQ) model
includes a ss-$SO_4^{2-}$ component estimating it to be 7% of the total SSA emissions. The mass fraction of ss-$SO_4^{2-}$ can be
estimated to be 0.25 of the Na$^+$ mass fraction *(Kelly et al., 2010; Neumann et al., 2016)* and applied in WRF-Chem to calculate
ss-$SO_4^{2-}$. Note that the total fraction of Na$^+$, Cl$^-$, marine organics and ss-$SO_4^{2-}$ is equal to 1.0, and additional mass is not added.
The mass fraction of ss-$SO_4^{2-}$ is estimated to be 9.9% of the total SSA emissions in our simulations.

### 4.6 Discussion

Average differences in aerosol concentrations between the HEM_NEW and CONTROL simulations are shown in Figures 5 and
6 for January and February 2014 super-micron and sub-micron aerosols, respectively. The updated model simulates less super-
micron Na$^+$ by up to 20 $\mu$ g m$^{-3}$, and Cl$^-$ by up to 30 $\mu$ g m$^{-3}$, especially south of Alaska and north of the Atlantic Ocean (Fig.
5). These decreases lead to an overall decrease (up to 2.5 $\mu$ g m$^{-3}$) in super-micron $NO_3^-$, over continental and coastal regions
and the North Atlantic. This is in agreement with *Chen et al. (2016)* who examined the influence of SSA on $NO_3^-$ and reported
that overestimation of SSA can lead to an overestimation of super-micron $NO_3^-$, due to formation of $NO_3^-$ via heterogeneous
uptake of nitric acid (HNO$_3$) on SSA. Furthermore, due to the addition of ss-$SO_4^{2-}$ component in the model, there is more
super-micron $SO_4^{2-}$, of up to 2 $\mu$ g m$^{-3}$, over marine regions. Super-micron $NH_4^+$ also increases (by up to 0.2 $\mu$gm$^{-3}$) in regions,
such as Siberia and North of Europe, coinciding with decreases and increases in $NO_3^-$ and $SO_4^{2-}$, respectively. Super-micron
OA increases by up to 0.6 $\mu$ g m$^{-3}$ due to the inclusion of marine organics. During winter, the Beaufort Sea, located north of
Alaska is covered by sea-ice. Here, the implemented changes lead to smaller decreases in super-micron Na$^+$ and Cl$^-$ compared
to ice-free regions such as the Aleutians islands, e.g., Simeonof site (Fig. 2a) further south. The local effect of sea-ice fraction
and open leads on SSA production is examined further in 5.4.

On the other hand, the effect of model updates on sub-micron Na$^+$ is smaller, with decreases of up to 0.25 $\mu$ g m$^{-3}$ south of
Alaska and the North Atlantic (Fig. 6) due to use of lower wind speed dependence (SALI14 instead of MO80). The lifetime
of SSA, estimated to be between 1 to 4 days over open ocean, in the Arctic and during wintertime *(Rhodes et al., 2017;*
*Xu et al., 2016; Huang and Jaeglé, 2017; Hoppel et al., 2002)*, could explain the small decrease of sub-micron Cl$^-$ over
continental coastal areas (e.g. south of Alaska) in HEM_NEW. This could also affect long range transport of sub-micron SSA



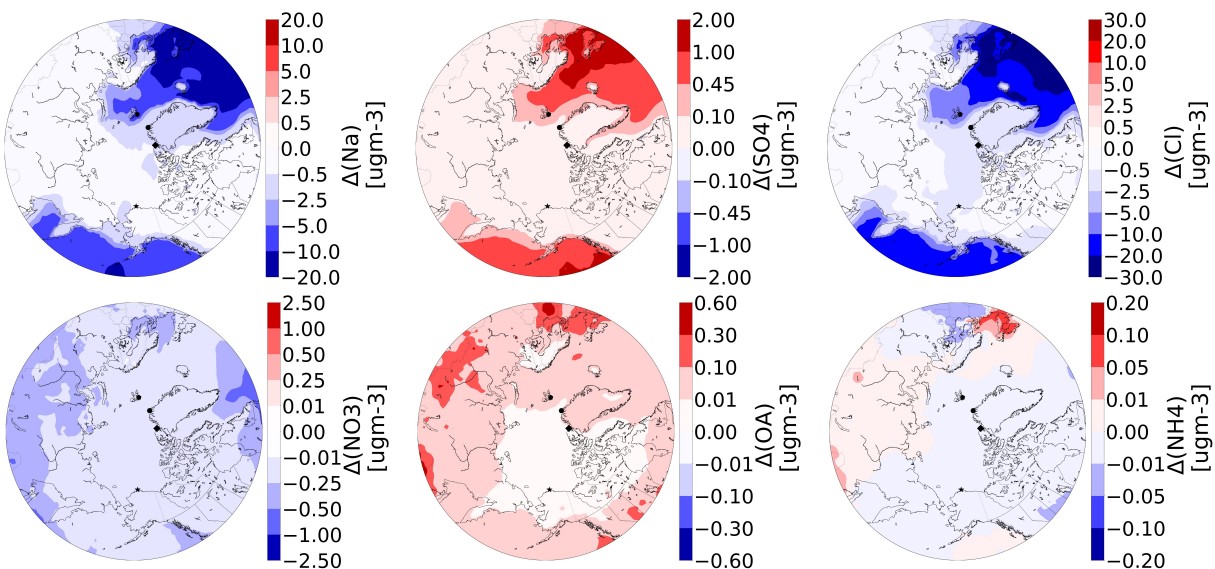

**Figure 5.** Average differences in super-micron aerosol mass concentrations ($\mu$ g m$^{-3}$) at the surface between HEM_NEW and CONTROL during January and February 2014. The black star in northern Alaska shows where Utqiaġvik is located. The black circle shows Alert, Canada, the black diamond shows Villum in Greenland, while the black pentagon shows Zeppelin, Svalbard.

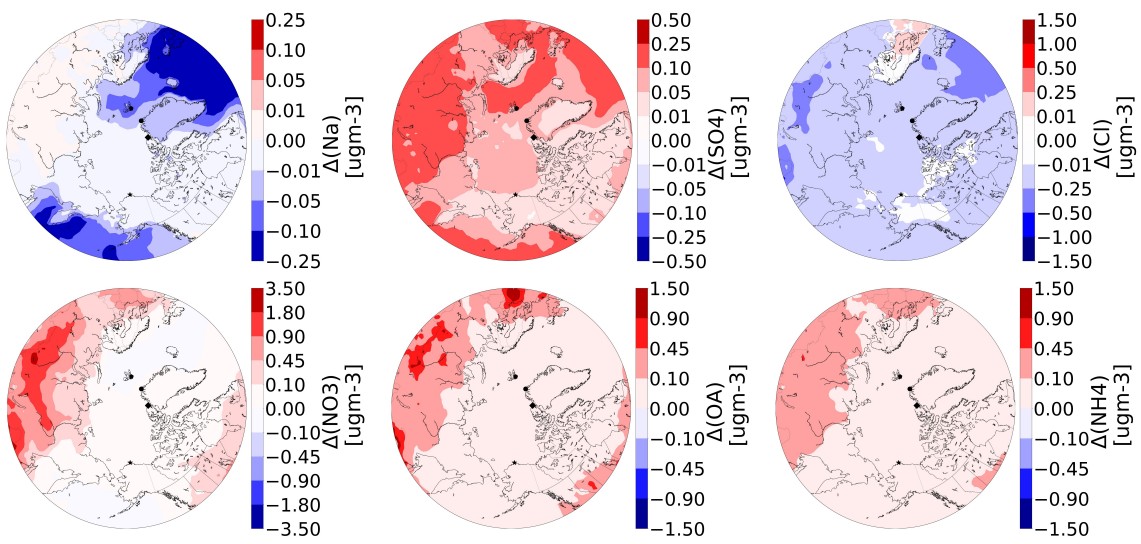

**Figure 6.** Average differences in sub-micron aerosol mass concentrations ($\mu$ g m$^{-3}$) and at the surface between HEM_NEW and CONTROL during January and February 2014. The black star in northern Alaska shows where Utqiaġvik is located. The black circle shows Alert, Canada, the black diamond shows Villum in Greenland, while the black pentagon shows Zeppelin, Svalbard.





from oceanic regions leading to decreases over continental regions, such as northeast United States of America (USA) and Siberia. Sub-micron OA increases by up to 1.5 $\mu$ g m$^{-3}$ due to inclusion of the F10 parametrization. Note that including ss-SO$_4^{2-}$ leads to a decrease in Na$^+$ and Cl$^-$ fractions per bin since no additional mass is added. In contrast to super-micron NO$_3^-$,

sub-micron NO$_3^-$ increases by 3.5 $\mu$gm$^{-3}$ over sources regions and total SO$_4^{2-}$ increases due to ss-SO$_4^{2-}$ component. Also, sub-micron NH$_4^+$ slightly increases, showing similar patterns to sub-micron NO$_3^-$ and SO$_4^{2-}$, probably due to a potential shift in the balance between (NH$_4$)$_2$SO$_4$ and NH$_4$NO$_3$.

To investigate the variations in modelled NO$_3^-$, SO$_4^{2-}$ and NH$_4^+$, the mean neutralized factor (f) is calculated (not shown here) as the ratio of NH$_4^+$ to the sum of (2SO$_4^{2-}$ + NO$_3^-$), in molar concentrations, following, for example *Fisher et al. (2011)*,

for sites in the Arctic with available observations of these aerosols. When f is equal to 1 aerosols are more neutralized, while when f < 1 then aerosols are acidic, and more acidic when f is closer to 0 *(Fisher et al., 2011)*. At all sites, except Zeppelin, higher molar concentrations were observed for SO$_4^{2-}$ compared to NO$_3^-$ and NH$_4^+$. At Utqiaġvik, the average observed value of f is equal to 0.15 for super-micron aerosols, whilst in the model f decreases from 0.7 to 0.66. This implies that observed super-micron aerosols are more acidic while in the model they are more neutralized *(Fisher et al., 2011)*, probably because

modelled NH$_4^+$ decreased more than SO$_4^{2-}$ and NO$_3^-$ (Fig. 4a) between the two simulations. There is less super-micron NH$_4^+$ in the model than the sum 2SO$_4^{2-}$ + NO$_3^-$, as in observations, however observed 2SO$_4^{2-}$ + NO$_3^-$ is much higher than modelled. Observed super-micron aerosols at Utqiaġvik are more acidic compared to sub-micron aerosols for which f equals 0.34. For sub-micron aerosols, HEM_NEW has an average f value of 0.08 compared to 0.01 in the CONTROL run. The increase in modelled sub-micron f could be due to the bigger increase in modelled NH$_4^+$ between the two simulations (Fig. 4b). However,

in the observations, the higher sub-micron f is because the sum 2SO$_4^{2-}$ + NO$_3^-$ is much higher than NH$_4^+$. At Alert (coarse mode), model f increases from 0.14 (CONTROL) to 0.19 (HEM_NEW), with observed f equal to 0.21, implying that model and observations are acidic, in contrast to Utqiaġvik modelled super-micron aerosols. Similar values of f are found for Zeppelin (coarse mode) and Villum (TSP) (0.12 for CONTROL, 0.13 and 0.18 for HEM_NEW, respectively) with observed aerosols (0.34 and 0.36 respectively) being less acidic at these sites. Overall the model inorganic aerosols are mostly too acidic compared

to the observations. This could be due to underestimation of anthropogenic sources of NH$_3$ on the above sites, originating from mid-latitudes. It can be noted that in the model is assumed that all of the aerosol species are internally mixed. However, in reality some of the NO$_3^-$ and SO$_4^{2-}$ are observed to be mixed with SSA (KRP18). Based on that, the calculated f for observations would be biased (too acidic), as some of the NO$_3^-$ and SO$_4^{2-}$ are present as Na$_2$SO$_4$ and NaNO$_3$.

Figures 2 and 3 show the effect of all the modifications (HEM_NEW) compared to CONTROL and the observations at four

Arctic and one sub-Arctic sites. At the two remote high-Arctic sites surrounded by sea-ice (Alert and Villum, Figure 2a,b), HEM_NEW captures better Na$^+$ and Cl$^-$ variability, with a small overestimation at Villum (maximum 0.2 $\mu$ g m$^{-3}$). Biases, in $\mu$ g m$^{-3}$, at Alert for Na$^+$ and Cl$^-$ decrease from 0.81 to 0.12 and from 1.05 to -0.03, respectively. Model results also improve at Villum for Na$^+$ and Cl$^-$ with biases reduced from 1.3 to 0.25 and from 1.9 to 0.22 $\mu$ g m$^{-3}$, respectively. The high variability in SSA at Villum at the end of January and the middle of February is likely to be due to fluctuations in sea-ice fraction around the

site, as seen in the FNL sea-ice reanalysis product (varies between 0.93 and 1.0-fully covered, in January and February). Also, HEM_NEW captures better NO$_3^-$ and NH$_4^+$ at Alert while slightly overestimates total SO$_4^{2-}$ (see APPENDIX C, Table C1). At



Villum, HEM_NEW captures better $SO_4^{2-}$ compared to CONTROL run, slightly underestimates $NH_4^+$ end of February, but still overestimates $NO_3^-$ (see APPENDIX C, Table C2). Similar results are found for Zeppelin where HEM_NEW simulates better $Na^+$, Cl⁻, $NO_3^-$ and $NH_4^+$, but overestimates $SO_4^{2-}$. More specifically, HEM_NEW slightly underestimates observed $Na^+$, Cl⁻

and $NH_4^+$, but the model results improve in this site. SSA updates also improve modelled $NO_3^-$ (see APPENDIX C, Table C3).

At Simeonof, HEM_NEW captures better $Na^+$, Cl⁻ and $NO_3^-$ variability during winter 2014 and, due to the inclusion of marine organics, the model simulates more tOC, although it still underestimates observed variability. Calculated biases decrease from 1.4 to 0.3, 2.0 to 0.1, 0.12 to 0.08, -0.08 to -0.05 $\mu$ g m⁻³ for $Na^+$, Cl⁻, $NO_3^-$ and tOC, respectively. Also, the addition of ss-$SO_4^{2-}$ in HEM_NEW leads to improvements (biases, RMSEs) in simulated $SO_4^{2-}$ even if the model occasionally underes-

timates by up to 0.6 $\mu$ g m⁻³. Similar patterns are found for the Gates of the Arctic in northern Alaska. $Na^+$ and Cl⁻ are lower in HEM_NEW while modelled $NO_3^-$ and tOC also improve, with biases decreasing for all the four species (0.56 to 0.16, 0.70 to 0.09, 0.26 to 0.18 and -0.24 to -0.21 $\mu$ g m⁻³ for $Na^+$, Cl⁻, $NO_3^-$ and tOC, respectively, see also APPENDIX C for RMSEs). HEM_NEW simulates more total $SO_4^{2-}$ at this site but still underestimates the observations, in particular nss-$SO_4^{2-}$. Here, the contribution of ss-$SO_4^{2-}$ is minimal, as shown in Fig. 3b. Thus, the model underestimation could be due to issues related to

long-range transport of nss-$SO_4^{2-}$, such as wet deposition, or to missing local anthropogenic sources (e.g. Prudhoe Bay oilfields). Additional wintertime production of $SO_4^{2-}$ via mechanisms not requiring sunlight may also contribute. For example, *McCabe et al. (2006)* suggested that there is secondary $SO_4^{2-}$ at Alert during wintertime from metal catalyzed $O_2$ oxidation of S(IV) (10–18%). Results from HEM_NEW also underestimate tOC at Gates of the Arctic, possibly due to underestimation of marine organics (see discussion in next section) or missing regional or remote sources.

Figure 4 compares results from CONTROL and HEM_NEW with observations for super-micron (weekly averages) and sub-micron (daily averages) aerosols at Utqiaġvik. While CONTROL overestimates SSA and $NO_3^-$ and underestimates $SO_4^{2-}$ (only non-ss-$SO_4^{2-}$), in general, HEM_NEW captures better observed super-micron $Na^+$, Cl⁻, $NO_3^-$ and $NH_4^+$ aerosols during the simulation period (Fig. 4a) (see also Appendix C). The $Na^+$ bias decreases from 0.3 to -0.07 $\mu$ g m⁻³ but Cl⁻ is now underestimated (bias decreases from 0.27 to -0.26 $\mu$ g m⁻³), due to the introduction of the SST dependence (not shown).

Also, there is more super-micron $SO_4^{2-}$ in HEM_NEW and the model slightly underestimates observed $SO_4^{2-}$ by about 0.1 $\mu$gm⁻³. Super-micron OA is smaller in magnitude compared to the other aerosol components. However, super-micron OA mass concentration measurements are not available in winter 2014 to evaluate the model. Overall, modelled super-micron SSA concentrations decrease in HEM_NEW, as at other remote sites (Fig. 2 & 3) in better agreement with the observations compared to the CONTROL run.

On the other hand, while HEM_NEW (Fig. 4b) represents better periods with low concentrations of sub-micron $Na^+$ and Cl⁻ at Utqiaġvik in January and February 2014 (up to 0.3 $\mu$ g m⁻³), it still underestimates episodes with very high observed $Na^+$ and Cl⁻, especially at the end of February 2014. The model simulates better $NO_3^-$ but underestimates $NH_4^+$ and $SO_4^{2-}$, especially at the beginning of January and end of February 2014. Sub-micron OA at 100 km ranges between 0.01 and 0.15 $\mu$ g m⁻³. However, observations of OA at Utqiaġvik during this period are not available with which to validate the model.

*Barrett et al. (2015)* and *Barrett and Sheesley (2017)* showed that OC at Utqiaġvik is influenced by primary and secondary biogenic carbon and fossil fuel carbon, with air masses originating from the Arctic Ocean, Russian and Canadian Arctic. More





**Table 2.** Calculated fractions of observed and modelled (HEM_NEW) SSA to total aerosol mass concentrations (summed from available observations at each site). For each site SSA are defined as the sum of Na$^+$, Cl$^-$ and ss-SO$_4^{2-}$. Total is defined as the sum of SSA and inorganic aerosols. Inorganic is the sum of nss-SO$_4^{2-}$, NH$_4^+$ and NO$_3^-$ for each station except for Simeonof and Gates of the Arctic where inorganic is the sum of nss-SO$_4^{2-}$ and NO$_3^-$. Note that NH$_4^+$ is rarely internally mixed within SSA aerosol, because most NO$_3^-$ and SO$_4^{2-}$ forms via Cl$^-$ (e.g. NaCl + HNO$_3$ -> NaNO$_3$ + HCl). Total_all below is defined as the sum of SSA, nss-SO$_4^{2-}$, NH$_4^+$, NO$_3^-$, BC, OA and dust (model only). The aerosol size for SSA, Total and Total_all varies per station and corresponds to observed aerosol sizes as described in Section 3.

| Sites | SSA/Total [obs] | SSA/Total [HEM_NEW] | SSA/Total_all [HEM_NEW] |
|---|---|---|---|
| Simeonof (fine mode) | 0.73 | 0.84 | 0.74 |
| Gates of the Arctic (fine mode) | 0.20 | 0.44 | 0.33 |
| Utqiaġvik-sub-micron | 0.60 | 0.22 | 0.13 |
| Utqiaġvik-super-micron | 0.93 | 0.57 | 0.54 |
| Alert (coarse mode) | 0.59 | 0.54 | 0.45 |
| Villum (TSP) | 0.32 | 0.63 | 0.52 |
| Zeppelin (coarse mode) | 0.56 | 0.75 | 0.62 |

specifically, *Barrett and Sheesley (2017)* made measurements of OC (diameter less than 10 $\mu$m) collected during winter 2012-2013 northeast of Utqiaġvik and reported average OC of 0.22 $\mu$ g m$^{-3}$. To compare directly with the model results we divide the modeled value by 1.4 (Fig. 5). In that case, modelled super-micron tOC at Utqiaġvik is three times less than the observed

OC, showing that the model lack sources of OC. *Shaw et al. (2010)* reported sub-micron OM equal to 0.43 $\mu$ g m$^{-3}$ during winter 2008 (November to February) at Utqiaġvik, almost double the simulated OA at Utqiaġvik. Their analysis showed that OM was correlated with organic and inorganic seawater components with the air masses originating along the coastal regions of the Northwest Territories of Canada. Also, the model results can be compared with weekly average sub-micron OM data collected at Alert (*Leaitch et al. (2018)* and Fig. 2). At Alert, OM reaches up to 0.25 $\mu$ g m$^{-3}$ in February 2014, which is almost

double compared to the model results for Utqiaġvik, North Slope of Alaska and Alert (Fig. 6). At Villum, a recent study by *Nielsen et al. (2019)* showed that OA peaks at 2.2 $\mu$ g m$^{-3}$ at the beginning (21 to 28 February 2015) of their study period. Their study shows that the majority of OA is mostly due to Arctic Haze influence (up to 1.1 $\mu$ g m$^{-3}$) with secondary influence, due to hydrocarbon-like organics (up to 1.0 $\mu$ g m$^{-3}$) and a marine influence (up to 0.2 $\mu$ g m$^{-3}$). Reasons for these differences on modelled and observed OA are investigated in the next section focusing on regional processes affecting SSA near northern

Alaska.

     Previous studies *(Quinn et al., 2002; Quinn and Bates, 2005; May et al., 2016; Kirpes et al., 2018, 2019)*, pointed out that SSA are an important contributor in the total sub-micron and super-micron mass fraction in the Arctic during wintertime. A recent study by *Moschos et al. (2022)* showed that during wintertime SSA dominates PM$_{10}$ (particulate matter with diameter $\leq$ 10 $\mu$m) mass concentrations at remote Arctic sites, including Alert (56%), Baranova (41%) (Russia), Gruvebadet (44%)

(Norway), Utqiaġvik (66%), Villum (32%), and Zeppelin (65%). In contrast, at sites such as Tiksi (Russia) and Pallas (Finland), SO$_4^{2-}$ and OA dominate (70% and 55%, respectively). To investigate the contribution of SSA to total mass concentrations





during the period of this study, the observed and modelled fraction of SSA to "total" (SSA plus inorganic) aerosols are estimated (see Table 2). However, it should be noted that this fraction varies between sites since not all components were measured.

Overall, taking into account the observations available at each site, the fraction of SSA to total SSA+inorganics is higher at
all the coastal sites (Utqiaġvik, Alert, Simeonof, Villum) and Zeppelin ranging from 54 to 93%. Only at the Gates of the Arctic and Villum stations the fraction of SSA is smaller (20% and 32%). The modelled HEM_NEW SSA fraction shows similar patterns (fraction ranges between 44% and 84%) compared to the observations. An exception is sub-micron modelled SSA at Utqiaġvik due to low modelled concentrations. When taking into account all aerosol components in the model, including OA, BC and dust, SSA is dominant at Simeonof, Utqiaġvik (super-micron), Zeppelin and Villum (more than 54%), whereas
at Alert, SSA contributes about 45%. This analysis shows that SSA is an important fraction of total fine mode, super-micron, coarse mode and TSP aerosols in the most Arctic coastal sites during wintertime.

## 5   Regional processes influencing SSA over northern Alaska

In this section, processes which could affect SSA emissions on a regional scale over northern Alaska are examined. In general, the improved model simulates better observed super-micron, TSP, fine and coarse mode SSA, $NO_3^-$ and $SO_4^{2-}$ aerosols at
different sites in the Arctic but the model has difficulties capturing sub-micron SSA during wintertime at at Utqiaġvik. Possible reasons for these discrepancies are investigated in model runs at 20 km resolution during the campaign periods in January and February 2014 with boundary and initial conditions from HEM_NEW. The sensitivity of modelled SSA to various processes is examined including aerosol dry deposition over snow/ice, inclusion of local marine organic aerosols, higher wind speed dependence for sub-micron SSA and representation of sea-ice fraction. The possible role of blowing snow and frost flowers is
also discussed. Details about the simulations are provided in Table 3.

**Table 3.** WRF-Chem model simulations including details about SSA treatments in the regional runs.

| Simulation Name | Description |
|---|---|
| Regional simulations [20km] | |
| ALASKA_CONTROL_FEB | Control run for February 2014 |
| DRYDEP_FEB | + Updated dry deposition velocities over snow-ice and |
| | open water on *(Zhang et al., 2001)* based on *(Nilsson et al., 2001)* |
| LOC_ORG_FEB | + Local source marine organics *(Kirpes et al., 2019)* |
| SSA_WS_DEP_FEB | + Sub-micron SSA wind-speed dependence *(Russell et al., 2010)* |
| NEW_ALASKA_FEB | DRYDEP_FEB + LOC_ORG_FEB |
| | + SSA_TEST_FEB + ERA5 sea-ice fraction |
| ALASKA_CONTROL_JAN | Control run for January 2014 |
| | (same setup as ALASKA_CONTROL_FEB) |
| NEW_ALASKA_JAN | including all updates in NEW_ALASKA_FEB |



## 5.1 Aerosol dry deposition

Previous studies have shown the importance of including wet and dry removal treatments in models *(Witek et al., 2007; Eckhardt et al., 2015; Whaley et al., 2022). Sofiev et al. (2011)* estimated that dry deposition, including sedimentation, could contribute more than 50% to SSA removal, especially for super-micron SSA. JA11, using model treatments for dry deposition
from *Zhang et al. (2001)* over land and *Slinn (1982)* over ocean, reported that the loss of super-micron SSA is dominated by dry deposition. In the quasi-hemispheric simulations, dry deposition velocities are calculated in MOSAIC based on the *Binkowski and Shankar (1995)* parametrization. Here, the *Zhang et al. (2001)* scheme is applied over Alaska, in which the dry deposition velocities are calculated taking into account the different land categories, in contrast to MOSAIC scheme, which uses universal values for processes such as Brownian diffusion and Schmidt number *(Slinn and Slinn, 1980; Slinn, 1982). Zhang et al.*
*(2001)* has been used in previous studies, for example, by *Fisher et al. (2011)* and *Huang and Jaeglé (2017)*. These studies applied aerosol dry deposition velocities of $3.0 \times 10^{-4}$ m s$^{-1}$ over snow and ice surfaces for all aerosol diameters and the dry deposition velocity is calculated as a function of aerosol diameter. *Zhang et al. (2001)* includes detailed treatments of deposition processes, such as Brownian diffusion, impaction, interception, gravitational settling and particle rebound, which highly vary depending on land surface type. Certain parameters link to interception, such as collection efficiency by interception, or
impaction processes (e.g. Stokes number) over specific land use categories (such as ice/snow and open ocean), are calculated without considering the radius of surface collectors *(Giorgi, 1988)*, but using kinematic viscosity of air, gravitational settling velocity of particle, friction velocity *(Slinn, 1982; Seinfeld, 1986)*. Thus, dry deposition velocities over ice/snow and open ocean are set equal to $3.0 \times 10^{-4}$ and $1.9 \times 10^{-3}$ m s$^{-1}$, respectively, for both sub- and super-micron aerosols, following *Nilsson and Rannik (2001)*, who reported dry deposition velocity measurements from an Arctic Ocean expedition in 1999. In that way,
the influence of more realistic dry deposition velocities on SSA aerosols is examined during wintertime.

Figures 7 and 8 show the effect of this modification for sub- and super-micron SSA and NO$_3^-$, respectively (differences between DRYDEP_FEB and ALASKA_CONTROL_FEB runs). Sub-micron Na$^+$, OA and NO$_3^-$ decrease very slightly, whereas super-micron Na$^+$, OA and NO$_3^-$ increase by up to 0.6, 0.02 and 0.3 $\mu$ g m$^{-3}$, respectively, with the largest increase over sea-ice areas or regions with snow cover. These changes in modelled sub- and super- micron aerosols are due to differences between
the dry deposition velocities in the two schemes. Over model grids covered with snow or ice and open ocean MOSAIC dry deposition velocities are smaller (larger) for sub-micron (super-micron) in magnitude compared to reported velocities by *Nilsson and Rannik (2001)*. During wintertime over northern (in-land) Alaska, all the grid cells during the simulations are snow covered. Based on these results, and the fact that super-micron Na$^+$ and Cl$^-$ are slightly underestimated at 100 km and Utqiaġvik (see section 4.5), the following simulations use the observed dry deposition velocities reported by *Nilsson and Rannik (2001)*.
These results show that sub- and super-micron (mostly) SSA are sensitive to different dry deposition parametrization in WRF-Chem. To address potential uncertainties in dry removal treatments and their influence on SSA regionally, a series of sensitivity tests are also performed. Firstly, correct modelling of aerosol dry deposition depends on the ability of the model to capture the structure of the Arctic boundary layer including vertical temperatures and winds. Model results at 20 km and 100 km horizontal resolution are compared against hourly in-situ 2 m, 10 m temperatures and 10 m wind speeds or temperature



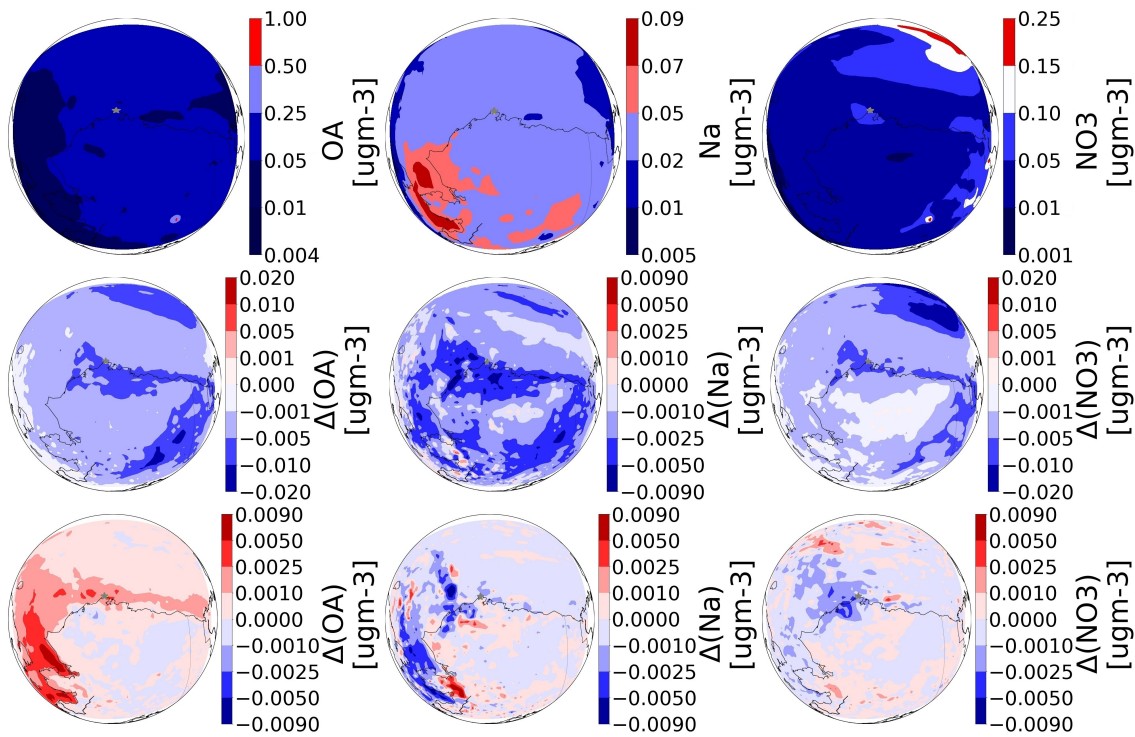

**Figure 7.** First row shows the average values of aerosol mass concentrations for sub-micron during February campaign. Average differences at the surface between DRY_DEP_FEB and ALASKA_CONTROL_FEB (second row), and between LOC_ORG_FEB and DRY_DEP_FEB (third row) during February campaign for sub-micron Na$^+$, OA, NO$_3^-$ ($\mu$ g m$^{-3}$). See text and Table 2 for detailed description of the model runs.

and wind speed profiles up to 4 km for January and February 2014 (see figures in APPENDIX D with calculated bias and RMSE for the two periods). Observed wind speeds during January ranged between 4.7 and 14.1 ms$^{-1}$ and wind directions were mostly easterly (77 to 135 degrees). During February, wind speeds ranged between 0.4 and 13.3 m s$^{-1}$ and wind directions were mostly easterly, except from 22 UTC 25 February to 11 UTC 26 February when the winds were westerly. In general, the model performs well at 20 km, and better than at 100 km, in terms of temperature and winds, although it slightly underestimates

observations at the surface. On the other hand, there are small discrepancies, of up to 10 degrees, between modelled (at 20 km) and observed wind direction at the Barrow site, near Utqiaġvik town, except at 26 February when these discrepancies are up to 70 degrees.

To examine further causes of variability in modelled dry removal of SSA, a sensitivity test is carried out where aerosol dry deposition and gravitational settling are switched off during a windy day, 28 February 2014. On this day, 10 m wind speeds at

Utqiaġvik varied between 7 and 13.5 m s$^{-1}$ and were easterly (104 to 130 degrees). This corresponds to a period when observed sub-micron Na$^+$ and Cl$^-$ concentrations were high, around 1.4 and 2.0 $\mu$ g m$^{-3}$, respectively (see Figure 10b). During this day the model captures quite well observed wind speeds and directions, with small differences of up to 2 m s$^{-1}$ and up to 10 degrees



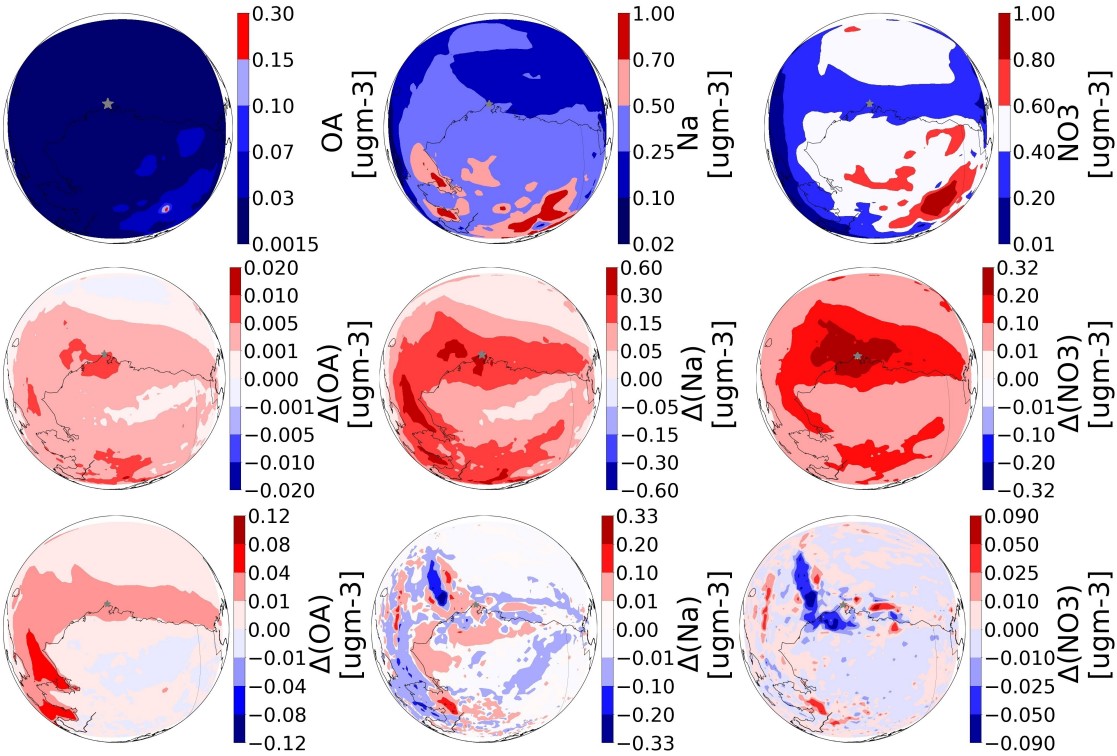

**Figure 8.** First row shows the average values of aerosol mass concentrations of super-micron during February campaign. Average differences at the surface between DRY_DEP_FEB and ALASKA_CONTROL_FEB (second row), and between LOC_ORG_FEB and DRY_DEP_FEB (third row) in super-micron Na$^+$, OA, NO$_3^-$ ($\mu$ g m$^{-3}$)during February campaign. See text and Table 2 for detailed description of the model runs.

differences in wind direction (not shown here). In this sensitivity run, the model simulates more super-micron SSA (an increase up to 0.8 $\mu$ g m$^{-3}$), which is expected due to the influence of gravitational settling on super-micron particles. The increase on sub-micron SSA is smaller. However, observations of dry deposition of different aerosols are needed to better constrain the model in the Arctic.

## 5.2 Local source of marine organics

For the simulations at 100 km, the F10 parametrization is used based on C:Chl-a from a cruise at mid-latitudes. Whilst phytoplankton blooms may not expected in the high Arctic winter, previous studies have shown evidence of sea ice biological activity under low light conditions in the Arctic *(Krembs et al., 2002; Lovejoy C., 2007; Hancke et al., 2018)*. In addition, *Russell et al. (2010)* (from now on RUS10) analysed samples from the International Chemistry Experiment in the Arctic Lower Troposphere (ICEALOT) cruise and found that most organic mass in clean regions of the North Atlantic and the Arctic is composed of carbohydrate-like compounds containing organic hydroxyl groups from primary ocean emissions. *Frossard et al.*





*(2014)* (FRSS14 from now on) investigated the sources and composition of atmospheric marine aerosol particles based on the
analysis of samples, including those from ICEALOT, reporting that ocean-derived organic particles include primary marine or-
ganic aerosols. In particular, they calculated the ratio of OC:Na$^+$ as a metric for comparing the composition of model-generated
primary marine aerosol and seawater, previously used by *Hoffman and Duce (1976)*, and reported OC:Na$^+$ ratios of 0.45 for
atmospheric marine aerosol particles. KRP19 also reported that during their campaigns in 2014 almost all the individual SSAs
had thick organic coatings (average C:Na mole ratios of 0.5 and 0.3 for sub-micron and super-micron SSA, respectively) made
up of marine saccharides. They also identified open sea ice leads enriched with exopolymeric substances as contributing to
organics in winter SSA.

Here, elemental fractions for sub- and super-micron aerosols sampled during the KRP19 campaigns are used to better con-
strain modelled organic marine emissions (mOC). More specifically, the ratio of sub- and super-micron OC:Na$^+$ is calculated,
following FRSS14 and using the elemental fractions from KRP19, as an indicator of the presence of a local source of marine
organics. The organic fraction of the total SSA for the high organic activity scenario in WRF-Chem is increased to 0.4 for sub-
micron (1st to 5th MOSAIC 8-bins) and to 0.11 for super-micron (6th to 8th MOSAIC 8-bins). Note again that no additional
SSA mass is added. Figures 7 & 8 show the sensitivity of the model results to including a larger marine organic fraction. Sub-
and super-micron OA concentrations increase on average by up to 0.009 and 0.12 $\mu$ g m$^{-3}$, respectively, especially south-west
of Alaska and along coastal areas, including Utqiaġvik. Sub-micron Na$^+$ and NO$_3^-$ slightly decrease (0.005 $\mu$ g m$^{-3}$) around
Utqiaġvik region, and super-micron Na$^+$ and NO$_3^-$ decrease north-west of Utqiaġvik.

KRP19 reported measured sub-micron organic carbon volume fractions based on analysis of 150 SSA particles between 0.3
$\mu$m and 0.6 $\mu$m comparable to organic carbon volume fractions observed in SSA produced in mid-latitude algal bloom regions.
This suggests the presence of significant organic carbon associated with locally produced SSA on the coast of northern Alaska.
There are two available daily observations at Gates of the Arctic during the February campaign. The model captures better
observed tOC at the end of February in the run LOC_ORG_FEB with higher organic fraction. However, it underestimates tOC
on 25 February when the observed tOC reached 0.33 $\mu$ g m$^{-3}$ (see APPENDIX F). As mentioned previously, this discrepancy
could also be due to missing local anthropogenic sources related to North Slope oil field emissions *(Gunsch et al., 2020)*.
In the following runs, marine organics based on the calculated ratio of OC:Na$^+$ are included instead of F10 considering the
importance of local SSA marine sources at Utqiaġvik (KRP19). By including a local source of marine organics in the model,
this leads to a better agreement with the findings of the previous studies discussed in section 4.6. It can be noted that there
are only sporadic measurements of OA/OC at remote Arctic sites and detailed long-term observations are not available which
might help to better constrain model simulations.

## 5.3 Wind-speed sensitivity to sub-micron SSA emissions

In the regional runs presented so far, the lower wind speed dependence based on satellite data is used (SALI14) since it
improves modelled SSA compared to observations in the 100 km runs (see in section 4.2). However, RUS10 found evidence
for higher wind speed dependence during the ICEALOT cruise in the Arctic. They found that wind speed is a good predictor of
a marine factor, calculated using positive matrix factorization, for sub-micron organic mass (OM1$_{sea}$). Their analysis showed a



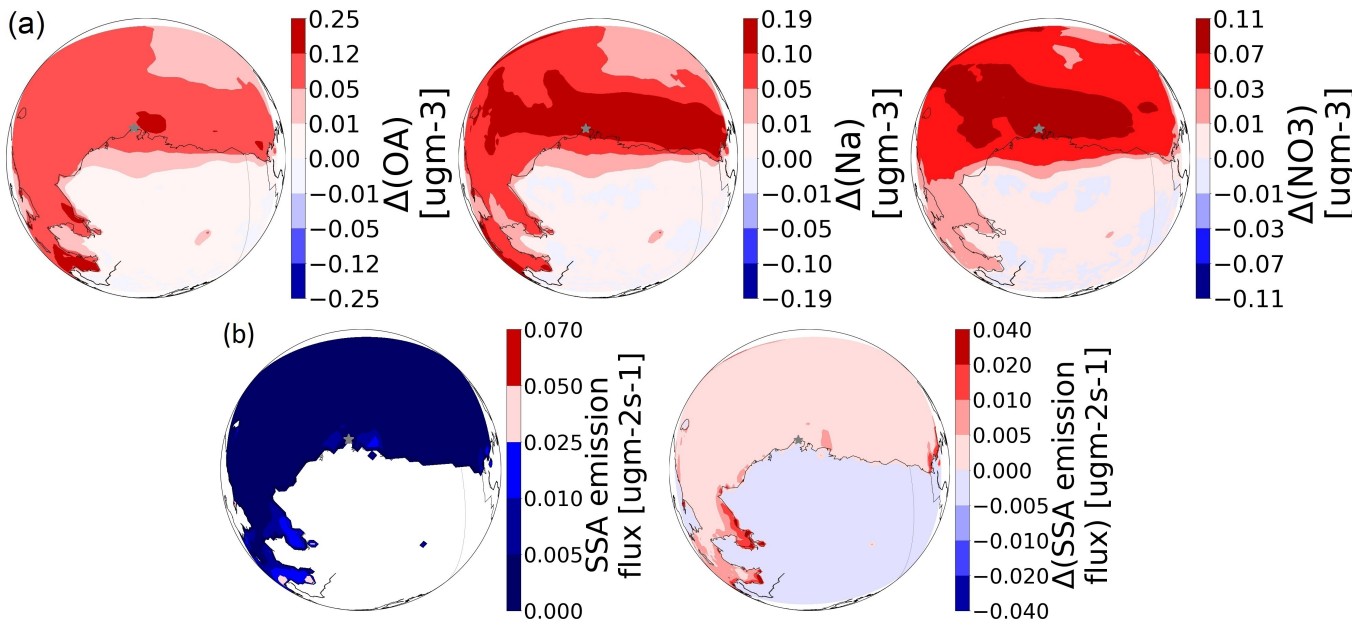

**Figure 9.** Average differences in mass concentrations of (a) sub- micron $Na^+$, OA, $NO_3^-$, in $\mu gm^{-3}$, at the surface between SSA_WS_DEP_FEB and LOC_ORG_FEB. Grey star indicates the location of Utqiaġvik. (b) The map on the left shows the average value of SSA emission fluxes in $\mu gm^{-2}s^{-1}$ during February campaign and the map on the right shows average differences between SSA_WS_DEP_FEB and LOC_ORG_FEB in $\mu gm^{-2}s^{-1}$.

high correlation between $OM1_{sea}$, sub-micron sodium ($Na^+1$) concentration and wind speed at 18 m (correlation r equal to 0.90 for the North Atlantic and Arctic region, see Table S3 Supplementary Material in RUS10). Average $OM1_{sea}$ concentrations
(0.2 $\mu$ g m$^{-3}$) reported by RUS10 for the eastern Arctic Ocean are about half those reported at Utqiaġvik by *Shaw et al. (2010)* during wintertime.

In a sensitivity simulation, the results from RUS10 are used to include a higher wind speed dependence for sub-micron SSA. Equations (5) and (6) from the RUS10 analysis for the Arctic legs of their cruise are applied to the model as a correction factor:

$$Na^+1 = 0.022 \times U_{18} - 0.012 \tag{4}$$

$$OM1_{sea} = 0.025 \times U_{18} - 0.049 \tag{5}$$

where $U_{18}$ is wind speed at 18 m in ms$^{-1}$, ranging between 2 and 14 m s$^{-1}$ (Figure 2, RUS10). RUS10 used $Na^+1$ as a proxy for sub-micron NaCl, and subsequently SSA, because $Na^+1$ equalled sub-micron $Cl^-1$ on a molar basis for the North Atlantic and Arctic sampling regions. Thus, Equation (5) is also used to estimate a correction factor for $Cl^-$. Here, wind speeds in the first model layer are used, i.e. around 26 m. Differences in $U_{18m}$ and $U_{26m}$ reach a maximum of 1 m s$^{-1}$ (see Fig.D1 in APPENDIX
D). Following RUS10, correction factors are only applied in the model to the number and mass of the SSA emissions when





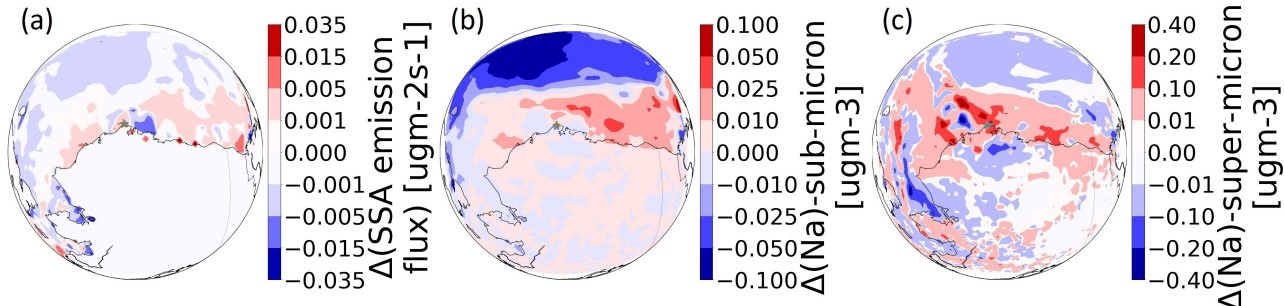

**Figure 10.** Average differences between ALASKA_NEW_FEB and SSA_WS_DEP_FEB showing the effect of switching from FNL to ERA5 sea-ice fractions during February for (a) SSA emission fluxes ($\mu$ g m$^{-3}$), (b) sub-micron mass concentration of Na$^+$ and (c) super-micron mass concentration of Na$^+$ in $\mu$ g m$^{-2}$ s$^{-1}$. The grey star shows the location of Utqiaġvik.

modelled wind speeds are between 2 and 14 ms$^{-1}$, and when RUS10-calculated sub-micron SSA concentrations are greater than model calculated SSA. In this way, SSA emissions are enhanced during periods of higher wind speeds.

To illustrate the sensitivity of the results to applying this correction, Fig. 9 shows sub-micron aerosol concentrations and SSA emission fluxes, the latter being the sum of dry mass emissions calculated in the model. Overall, this leads to an increase
of 0.25, 0.19 and 0.11 $\mu$ g m$^{-3}$ in sub-micron Na$^+$, NO$_3^-$ and OA, respectively, over the Utqiaġvik region and southwest Alaska during the February campaign. The SSA emission flux is influenced directly by the area in the model grid which is ice-free. This leads to SSA production east and west of Utqiaġvik while the highest values are southwest of Alaska. By adding the RUS10 correction, SSA emission fluxes increase slightly by up to 0.035 $\mu$ g m$^{-2}$ s$^{-1}$ along the southwest Alaskan coast, and by up to 0.015 $\mu$ g m$^{-2}$ s$^{-1}$ around Utqiaġvik. RUS10 showed that sub-micron SSA and wind speed are well correlated over
open ocean. Thus, a correction factor to sub-micron SSA, based on is-situ data, improves sub-micron model SSA and could be included in future simulations for studies focusing on the Arctic region.

### 5.4 Sea-ice fraction

The sensitivity of modelled SSA to prescribed sea-ice fractions during wintertime and the role of leads, is investigated since KRP19 already pointed out the importance of using realistic sea-ice to simulate marine aerosols. High spatial resolution images
of sea-ice cover are available, including during the Polar Night, from a marine radar operating on top of a building in downtown Utqiaġvik (71°17'13" N, 156°47'17" W), 22.5 m above sea level, with a range of up to 11 km to the northwest (http://seaice.alaska.edu/gi/data/barrow_radar) *(Druckenmiller et al., 2009; Eicken et al., 2011). May et al. (2016)* showed increased super-micron Na$^+$ mass concentrations during periods of elevated wind speeds and lead presence, in a multiyear study using the sea ice radar data at Utqiaġvik. Between 23-28 January 2014, when the winds at Barrow observatory were easterly, the radar
showed that the coastal area east of Utqiaġvik featured leads (KRP19). From 24-28 February 2014, the west coastal area also featured leads as shown by Moderate Resolution Imaging Spectroradiometer (MODIS) satellite images (KRP19). To examine the sensitivity of SSA emissions to sea-ice cover, ERA5 sea-ice fractions with a resolution of 0.25° x 0.25° are used instead




of FNL fraction at 1.0º x 1.0º resolution. Note that only sea-ice fraction field is different, while the rest of the meteorological fields are from FNL.

Results for February are shown in Fig. 10. The SSA emission flux (Fig. 10a) increases over a small region around Utqiaġvik and across the North Slope of Alaska due to decreased sea-ice fraction but decreases east of Utqiaġvik and southwest of Alaska (e.g. Selawik Lake and Norton Bay) due to increased sea-ice fraction. Sub-micron $Na^+$ slightly increases along the north coast of Alaska by up to 0.1 $\mu gm^{-3}$ and around Utqiaġvik (see Fig. 10b) and super-micron $Na^+$ increases by up to 0.4 $\mu$ g m$^{-3}$ around Utqiaġvik and decreases by up to 0.4 $\mu$ g m$^{-3}$ southwest of Alaska (Fig. 10c). The model results for January indicate that there

is less sea-ice in the region around Utqiaġvik and south west of Alaska.Therefore, higher SSA emission fluxes were simulated for February (0.035 $\mu$ g m$^{-2}$ s$^{-1}$) compared to January (0.015 $\mu$ g m$^{-2}$ s$^{-1}$) (maps not shown here).

    Two further simulations are performed to examine model sensitivity to sea-ice fraction. First, ERA5 sea-ice fractions are set equal to 0 (ice-free conditions) to the north, west, and east of Utqiaġvik to examine the effect of having ice-free conditions and the presence of open leads locally (as seen by the radar). Second, ERA5 sea-ice fractions are set equal to 0.75 north, west, east

of Utqiaġvik and northwest of Alaska. In both cases, the model is run on a windy day (28 February 2014). The first sensitivity test leads to an increase in SSA emission fluxes by up to 0.2 $\mu$ g m$^{-2}$ s$^{-1}$ where sea-ice fraction equals zero (not shown) and to an increase of up to 1.2 $\mu$ g m$^{-3}$ and 0.05 $\mu$ g m$^{-3}$ in super-micron and sub-micron $Na^+$ respectively. The second sensitivity test yields similar results. This is because ERA5 sea-ice fractions are higher than the test case (0.75) leading to an overall increase in the SSA emission flux of up to 0.02 $\mu$ g m$^{-2}$ s$^{-1}$, especially east of Utqiaġvik, affecting primarily super-micron SSA

(increases of up to 1.5 $\mu$ g  m$^{-3}$) rather than sub-micron SSA, probably due to the short simulation period.

    These results illustrate the sensitivity of super-micron SSA, in particular, to the prescribed sea-ice fraction and point out the need of improving this in models. Regarding sub-micron SSA, which is less sensitive to local sea-ice in these model simulations, there is the possibility that missing mechanisms influencing sub-micron SSA emissions need to be included such as SSA production from breaking waves in the surf zone for particles with diameters between 1.6 and 20 $\mu$m *(De Leeuw et al.,*

*2000)* or diameters ranging between 0.01-0.132 (ultrafine), 0.132-1.2 and 1.2-8.0 $\mu$m *(Clarke et al., 2006)*, which would be important and in the ice free ocean.

## 5.5   Evaluation against observations in northern Alaska

The model is also run for January 2014 including all the updates described above (see Table 2 and section 2.3). Fig. 11 shows the comparison between runs with and without all these updates compared to sub-micron aerosol observations at Utqiaġvik

during the January and February campaigns. Note that the focus of this section is on sub-micron SSA, as there are not detailed super-micron observations during the periods of the simulations due to their weekly temporal variation and due to the fact that the model still underestimates observed sub-micron SSA at Utqiaġvik.

    There are differences in the observations between the two periods. While sub-micron observed $Na^+$ and $Cl^-$ did not exceed 1 $\mu$ g m$^{-3}$ during January, observed sub-micron $Na^+$ and $Cl^-$ concentrations reached up to 2.5 $\mu gm^{-3}$ in February. As noted earlier

such high concentrations of $Na^+$ and $Cl^-$ were not observed at Alert and Villum during January and February 2014. This could be explained by the fact that these two sites are entirely surrounded by more sea-ice in winter. Overall, the model simulates





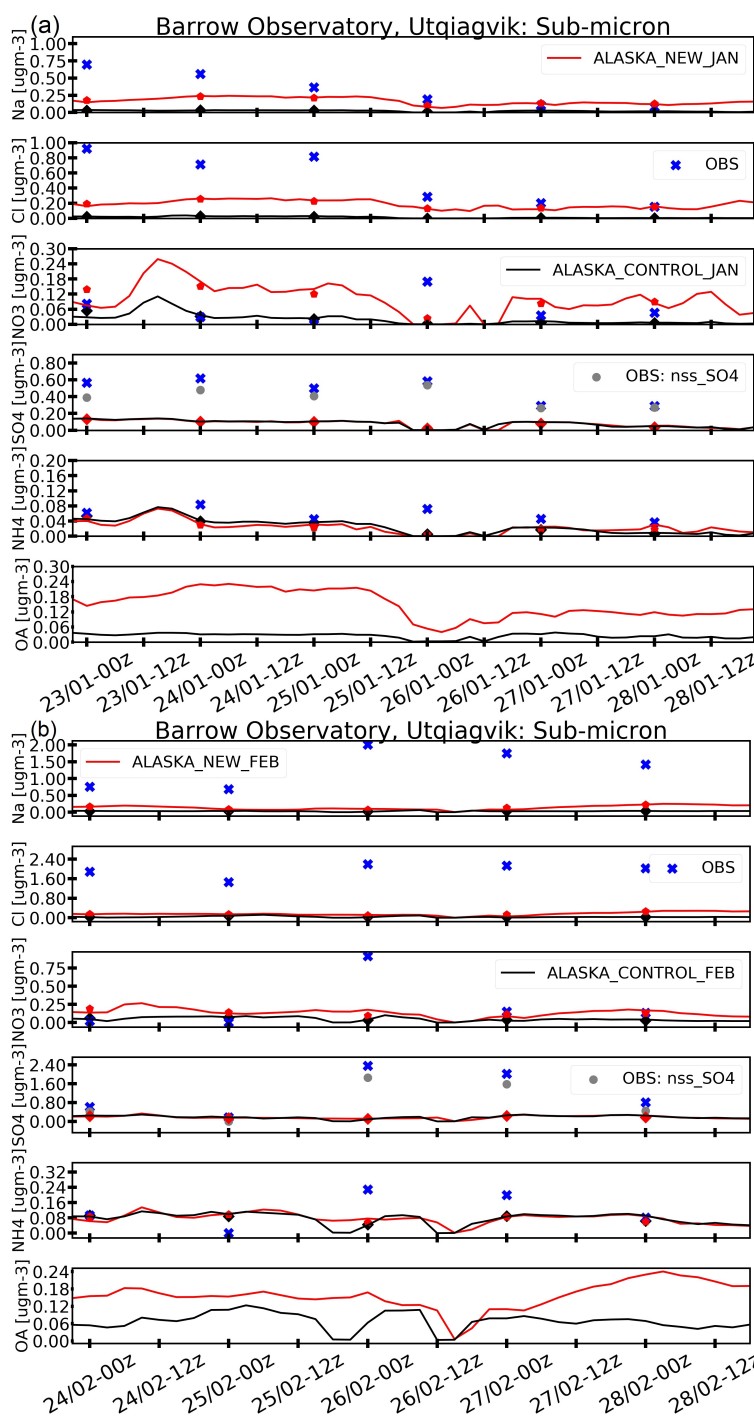

**Figure 11.** Time-series during a) January and b) February 2014 of sub-micron mass concentrations of $Na^+$, $Cl^-$, $NO_3^-$, $NH_4^+$, $SO_4^{2-}$, in $\mu$ g m$^{-3}$, simulation period. Model simulations are validated against in-situ sub-micron aerosols at Utqiaġvik, Alaska, in UTC (every 12h; 00z,12z). The black line shows model results from the CONTROL run; the red line shows the ALASKA_NEW run, while the daily observations are shown in blue crosses. The corresponding model daily averages are shown as black diamonds for the control simulation and as red pentagons for the ALASKA_NEW runs. See the text for details about the observations and model runs.





better observed sub-micron Na[+] and Cl[-] in January but still underestimates concentrations by up to 0.3 and 0.6 $\mu$ g m[-3], respectively, while sub-micron $NO_3^-$ is slightly overestimated. Biases in January decrease from -0.31 to -0.16, -0.50 to -0.33 and -0.04 to 0.039 $\mu$ g m[-3], respectively. On the other hand, sub-micron Na[+], Cl[-] are still underestimated in the run including all the updates (ALASKA_NEW_FEB) by up to 2.0 $\mu$ g m[-3] in February indicating that there are missing processes in the model linked to sub-micron SSA emissions, as discussed earlier. However, overall the results at Utqiaġvik in February, including all the updates (ALASKA_NEW_FEB), are better compared to the control simulation (ALASKA_CONTROL_FEB). Biases for Na[+], Cl[-] and $NO_3^-$ decrease from -1.29 to -1.18, -1.90 to -1.78 and -0.20 to -0.11 $\mu$ g m[-3], respectively. During both months, the model lacks $SO_4^{2-}$ due to missing local anthropogenic sources, as discussed in section 4.6 and due to small contribution from ss-$SO_4^{2-}$ as is shown also in section 4.1 for different Arctic sites. Missing aqueous phase reactions, such as the oxidation of $SO_2$ by ozone in alkaline SSA aerosols *(Alexander et al., 2005)* and $SO_4^{2-}$ production from metal catalyzed $O_2$ oxidation *(McCabe et al., 2006)* are missing from the model and might explain these high discrepancies compared to sub-micron observations at Utqiaġvik. Also, the variations in modelled $NH_4^+$ between the ALASKA_CONTROL and ALASKA_NEW for January and February simulations are small. The model underestimates observed $NH_4^+$, which peaks at 0.2 $\mu$ g m[-3] during February. Calculated biases and RMSEs for all aerosol species and for January and February campaigns are given in APPENDIX F.

Comparison with data from Gates of the Arctic (see APPENDIX E) shows that there are not significant differences between the control run and including all the updates in February 2014. The model still underestimates observed tOC due to missing local anthropogenic sources and overestimates $SO_4^{2-}$, $NO_3^-$, Na[+] and Cl[-]. However, due to short period of the simulation only two observations are available, thus more detailed observations are needed to examine further the reason why the model differs from the observations at this site.

## 5.6 Are blowing snow and/or frost flowers a source of sub-micron SSA during wintertime at Utqiaġvik?

Lastly, we consider whether enhanced SSA at Utqiaġvik could be due to blowing snow or frost flower sources. We noted earlier that KRP19 found no evidence of blowing snow or frost flowers at this site but that SSA originated from open leads during wintertime. The findings of KRP19 are supported by the earlier laboratory study of *Roscoe et al. (2011)* who reported that frost flowers are not an efficient source of SSA. However, an older study by *Shaw et al. (2010)* found that during winter at Utqiaġvik surface frost flowers formed on the sea and lake ice are a source of ocean derived OM. Modelling studies that have included a source of blowing snow and frost flowers suggest that they are contributing to SSA at this time of year at Utqiaġvik, Alert and Zeppelin *(Xu et al., 2013, 2016; Huang and Jaeglé, 2017; Rhodes et al., 2017)*.

To investigate whether blowing snow or frost flowers could also be a source of SSA during the campaigns at Utqiaġvik, depletion factors are estimated following *Frey et al. (2020)*. *Frey et al. (2020)* reported that blowing snow was the main source of SSA rather than frost flowers and open-leads in Antarctic wintertime, based on $SO_4^{2-}$ and Br[-] depletion in SSA being indicative of blowing snow origin, and not sea water. Here, depletion factors are calculated using modelled and observed sub-micron aerosol concentrations, during the campaign periods. More specifically, $SO_4^{2-}$ depletion relative to Na[+] ($DF_{SO_4^{2-}}$), Na[+] depletion relative to Cl[-] ($DF_{Na^+}$) and Br[-] depletion relative to Na[+] ($DF_{Br^-}$ only for observations in this case) are calculated using the following equation:



**Table 4.** Average sub-micron modelled and observed depletion factors, following *Frey et al. (2020)*, during the January and February campaigns 2014 in Utqiaġvik. Model results for ALASKA_NEW_JAN and ALASKA_NEW_FEB simulations are shown here, respectively. Observations refer to sub-micron data from NOAA. See text for details.

| Depletion Factors | Model | Observations |
|---|---|---|
| January campaign | | |
| $DF_{SO_4^{2-}}$ | -0.77 | -7.56 |
| $DF_{Na^+}$ | -1.05 | -0.09 |
| February campaign | | |
| $DF_{SO_4^{2-}}$ | -4.8 | -2.15 |
| $DF_{Na^+}$ | -1.1 | -0.19 |
| $DF_{Br^-}$ | - | 0.063 |

$$DF_x = 1 - \frac{R_{smpl}}{R_{RSW}} \qquad (6)$$

where, R is the mass ratio (x:y) in the model or in the sample (smpl) and in reference seawater (RSW) *(Millero et al., 2008)*. A depletion factor ($DF_x$) between 0 (small) and 1 (strong) indicates 0–100% depletion, whereas $DF_x$ smaller than 0 indicates enrichment. *Frey et al. (2020)* suggested, based on depletion of $SO_4^{2-}$ relative to $Na^+$, that most SSA originates from blowing
snow on sea-ice with minor contributions from frost flowers, and not from open leads.

Average values of modelled and observed DFs are shown in Table 4. In January, observed $SO_4^{2-}$ concentrations are 8.56 times more than in reference seawater, possibly due to internal mixing with anthropogenic $SO_4^{2-}$ from NSA oilfield emissions (KRP18), whilst in the model, $SO_4^{2-}$ concentrations are 1.77 times higher than in reference seawater, showing enrichment in both cases (Table 4). Modelled and observed depletion factors also show enrichment in February. This is in contrast with results
from *Frey et al. (2020)* who reported substantial depletion. They also reported a case of enrichment due to possible contamination from the ship, an anthropogenic source. Our modelling results and the observations at Utqiaġvik indicate enrichment of $SO_4^{2-}$ relative to $Na^+$, suggesting that blowing snow and frost flowers are not a source of SSA, at least during these campaigns. Previous studies *(Douglas et al., 2012; Jacobi et al., 2012)* suggested that blowing snow and frost flowers near Utqiaġvik are characterised by $SO_4^{2-}$ depletion compared to seawater . $Na^+$ depletion relative to $Cl^-$ during both campaigns also shows
enrichment, albeit more negligible in the observations than in the model. Observed $Na^+$ depletion relative to $Cl^-$ is 1.09 or 1.19 times more than in reference seawater, during January and February, respectively.

SSA can also play an important role in polar tropospheric ozone and halogen chemistry through the release of active bromine during spring *(Fan and Jacob, 1992; Simpson et al., 2007; Peterson et al., 2017)*. Reactions involving bromine are an important sink of ozone ($O_3$) *(Barrie, 1986; Barrie et al., 1988; Wang et al., 2019a; Marelle et al., 2021)* and also cause mercury oxidation
*(Schroeder et al., 1998)*. $Br^-$ depletion relative to $Na^+$ is calculated only during February, since observed $Br^-$ was zero during the period of January campaign, and indicates a small depletion in reference seawater. The calculated observed mass ratio of





**Table 5.** Average modelled and observed molar ratios for sub-micron SSA, following *Kirpes et al. (2019)*, during January and February campaign 2014 in Utqiaġvik. Model results from ALASKA_NEW_JAN and ALASKA_NEW_FEB simulations are used. Observations refer to sub-micron data from NOAA.

| Molar ratios | Model | Observations |
|---|---|---|
| January campaign | | |
| $SO_4^{2-}$:$Na^+$ | 0.11 | 0.55 |
| $Cl^-$:$Na^+$ | 0.74 | 1.1 |
| February campaign | | |
| $SO_4^{2-}$:$Na^+$ | 0.37 | 0.2 |
| $Cl^-$:$Na^+$ | 0.8 | 1.08 |

$Br^-$ to $Na^+$, based on the available observations of $Br^-$ during February, indicates a seawater origin. The observed mass ratio of $Br^-$ to $Na^+$ ranges between 0.0057 and 0.0059, while the mass ratio of $Br^-$ to $Na^+$ in reference seawater is equal to 0.006. On the other hand, *Frey et al. (2020)* reported no or little $Br^-$ depletion relative to $Na^+$ due to $Br^-$ loss at the surface and small

depletion further aloft. For a more comprehensive analysis, observations are required at different locations and altitudes across coastal northern Alaska.

We note that the version of WRF-Chem used in this study does not include halogen chemistry. It has since been implemented in a later version by Marelle et al. (2021) to examine ozone depletion events during March-April 2012 at Utqiaġvik. Heterogeneous reactions on sea salt aerosols emitted from the sublimation of lofted blowing snow were included. Their results

suggested that blowing snow could be a source of SSA during spring although it should be noted that this version of the model overestimated SSA at remote Arctic sites, such as Alert and Villum, when blowing snow was included as a source of SSA. Also, they did not examine wintertime conditions.

Finally, following KRP19, modelled and observed molar ratios of sub-micron $Cl^-$:$Na^+$ and $SO_4^{2-}$:$Na^+$ are estimated to further examine the origins of SSA and to compare our findings with KPR19 (see Table 5). Observed molar ratios of $Cl^-$:$Na^+$

and $SO_4^{2-}$:$Na^+$ for January and February campaign periods agree with KRP19 ($Cl^-$:$Na^+$ equal to 1.08, see KRP19 supplement - Table S3 and text). This indicates a seawater origin (following *Pilson (2012)*), and confirms the findings of KRP19 that there was no evidence for blowing snow and frost flowers as a source of SSA during the campaigns. Model averaged molar ratios are smaller in magnitude than the observations. Observed and modelled ratios differences in magnitude could be altered by the fact that the model underestimates sub-micron SSA and $SO_4^{2-}$, due to missing mechanisms for sub-micron SSA emissions

and local/regional anthropogenic sources of $SO_4^{2-}$. Differences between observed and modelled $Cl^-$:$Na^+$ ratios could also be due related to issues with modelled SSA lifetime and chemical processing during long range transport. Previous studies found that sub-micron SSA have larger chloride depletion than super-micron SSA *(Barrie et al., 1994; Hara et al., 2002; Leck et al., 2002)*. *May et al. (2016)* used molar ratio enrichment factors of $Cl^-$:$Na^+$ as an indicator of long-range transport influence on SSA at Utqiaġvik. They reported that $Cl^-$ depletion was larger for sub- than super-micron SSA due to a longer lifetime. On

average during the simulation periods in January and February 2014, the results indicate that modelled $Cl^-$ has undergone





significant atmospheric processing. This is consistent with KRP18 observing the presence of both nascent (locally-produced) SSA and aged (partially chloride-depleted) SSA. Based on this analysis of depletion factors and molar ratios, little evidence suggest a blowing snow influence on SSA during the campaigns at Utqiaġvik is found. Rather, the presence of predominantly easterly winds (s.s. 5.3) and the presence of leads east of Utqiaġvik (especially during February), suggests that the primary

source of SSA was marine from open-leads, in agreement with the findings of KRP19.

## 5.7 Conclusions

In this study WRF-Chem is used to investigate Arctic Haze composition at remote Arctic sites during wintertime with a particular focus on SSA, processes influencing SSA emissions and the contribution of SSA to Arctic Haze. Model performance is evaluated first in terms of reproducing aerosol composition in the Arctic before focusing on processes influencing SSA at

regional scales over northern Alaska during winter 2014.

The control version of WRF-Chem overestimates super-micron, coarse mode and TSP SSA due to missing and out of date SSA emission treatments in the model. In particular, the addition of a more realistic wind speed dependence for SSA, based on satellite data, and inclusion of a dependence of SSA emissions on SSTs leads to improved results for super-micron, coarse mode and TSP SSA and $NO_3^-$ over the Arctic. The latter has already been included in certain modelling studies. Also, recent data

analysis studies in the Arctic have pointed out that wind speed alone cannot predict SSA production and that other mechanisms, such as SST dependence, are needed. However, there are still uncertainties regarding the role of SSTs in SSA production. Other factors such as seawater composition, wave characteristics, fetch model and salinity need to be considered in future versions of WRF-Chem. In this study, marine organic aerosol emissions are also activated in the model since they are an important component of SSA in the Arctic and globally. Inclusion of all these updates leads to improved representation of SSA over the

wider Arctic. Modelled super-micron, coarse mode and TSP SSA are reduced at all Arctic sites in better agreement with the observations. Results for $NO_3^-$ are also improved overall due to less formation via heterogeneous uptake of $HNO_3$. Inclusion of the SST dependence only has a small effect on sub-micron SSA in the Arctic. In the future, other SST dependencies could be considered such as that proposed by *Sofiev et al. (2011)* which could increase sub-micron SSA at low temperatures *(Salter et al., 2014, 2015; Barthel et al., 2019)*. However, further field data studies are needed to confirm such dependencies in the

Arctic.

A source of ss-$SO_4^{2-}$ is also added to the model leading to improved modeled $SO_4^{2-}$ in the high Arctic (e.g. Alert) and Alaskan (e.g. Gates of the Arctic, Simeonof) sites. However, at sites such as Utqiaġvik, which may be influenced by the Prudhoe Bay oilfields, the model still underestimates sub-micron $SO_4^{2-}$ possibly due to missing anthropogenic emissions. Missing aqueous chemical formation of $SO_4^{2-}$ in dark conditions may also explain these discrepancies (e.g. $SO_4^{2-}$ production

from metal catalyzed $O_2$ oxidation of S(IV), *McCabe et al. (2006)*). Results from the improved quasi-hemispheric run indicate a shift in the balance between $(NH_4)_2SO_4$ and $NH_4NO_3$, with aerosols being less acidic than the base version of the model.

Overall, super-micron, coarse mode and TSP SSA, OA, $SO_4^{2-}$, $NH_4^+$ and $NO_3^-$ are improved in the HEM_NEW quasi-hemispheric simulation compared to observations at Arctic sites, based on biases and RMSEs. However, the model underestimates sub-micron SSA at Utqiaġvik where there are episodes with significantly higher SSA compared to other Arctic sites.





Model sensitivity to different processes affecting SSA over northern Alaska during winter is explored. KRP19 pointed out that there is sea ice biology influence at Utqiaġvik during wintertime and that marine emissions are an important source of organic aerosols at this location. In order to include local sources of marine organics, the ratio of OC:Na$^+$ is used leading to higher modelled OA, in better agreement with previous measurements at this site (and at Alert) and its advised to be included in future WRF-Chem simulations in the Arctic region. To further explore the uncertainties on sub-micron SSA, ERA5 sea-ice fraction is tested in the model. The results, in combination with different sensitivities changing sea-ice fraction, show that super-micron SSA are more sensitive to sea-ice treatments than sub-micron SSA in the model. The use of satellite sea-ice data, combined with higher resolution simulations over Utqiaġvik and coastal Arctic sites, will help to gain more detailed insights into the influence of open-leads on SSA production during wintertime. The results of this study also highlight that SSA dry removal is less important than the role of open leads in the Arctic during wintertime. The role of wet deposition on SSA is also examined. In that case, the precipitation flux is doubled and as result super-micron SSA decreased, but the sensitivity did not affect sub-micron SSA. Wet deposition is not addressed further in this study, because according to NOAA climate data recorded precipitation and snowfall was the lowest during February 2014. Wet deposition is addressed in details in the companion paper for BC. Our results suggest that further investigation is needed to determine more realistic dry deposition velocities over snow, ice and ocean in the Arctic and to derive more realistic sea-ice fractions, including the presence of open leads, which can vary over periods of days. The sensitivity of model results to using a higher wind speed dependence, based on data from *Russell et al. (2010)*, is investigated for sub-micron SSA. This leads to small improvements in the model sub-micron SSA, with the model performing better during January than February period of the campaign.

Further analysis is required to understand the origins of, in particular sub-micron SSA in northern Alaska, and to improve their representation in the WRF-Chem model. For example, missing sources of sub-micron SSA, such as a source function for ultrafine SSA particles due to breaking waves *(Clarke et al., 2006)* could be included. Also, anthropogenic sources of Cl$^-$ may need to be considered, such as road salt in urban areas *(McNamara et al., 2020; Denby et al., 2016)* or coal combustion, waste incineration, and industrial activities *(Wang et al., 2019b)* which are not included in current global inventories. The model also lacks anthropogenic emissions of Na$^+$. Anthropogenic sources of Na$^+$ could be wastewater and sewage treatment systems, contamination from landfills and salt storage areas (e.g. *Panno et al. (2006)*). However, detailed analysis of depletion factors and molar ratios at Utqiaġvik, Alaska showed that during the simulation period the main source of SSA are from marine emissions including open ocean or leads and there is no evidence of frost flowers or blowing snow as a source of SSA, at least during the periods considered in this study, in agreement with the findings of KRP19. Further observations from field measurements are needed to better understand SSA emissions and their dependencies.

This model study supports recent findings based on observations that SSA make an important contribution to super-micron (coarse mode, TSP) mass concentrations during wintertime at remote Arctic sites. Future work has to consider carefully possible sources of sub-micron SSA and their inclusion in models, in order to explain observed SSA during wintertime. Processes linked to the open ocean are likely to become more important with decreasing sea-ice cover in the Arctic due to climate warming. Observations of SSA components including organic aerosols (often missing) are needed to improve understanding about processes and their treatments in models, and in order to reduce uncertainties in estimation of aerosol radiative effects.



*Code availability.* Available upon request.

*Data availability.* All data used in the present paper for Zeppelin and Alert are open access and are available at the EBAS database infrastructure at NILU - Norwegian Institute for Air research: http://ebas.nilu.no/. Observations for Villum are obtained after personal communication with Henrik Skov. Observations from IMPROVE database can be obtained from: http://views.cira.colostate.edu/fed/QueryWizard/. Sub- and super-micron aerosol mass concentrations at Utqiaġvik, Alaska can be obtained from the follow link: https://saga.pmel.noaa.gov/data/
stations/.





**Table A1.** Land Surface model's (NOAH MP) parametrization. "Opt_" indicates the namelist option for NOAH MP.

| NOAH MP Parametrization | |
|---|---|
| Dynamic Vegetation (DVEG) | On |
| Stomatal Resistance | Ball-Berry *Ball et al. (1987)*, |
| | *Collatz et al. (1991)*, |
| | *Collatz et al. (1992)*, *Bonan (1996)*, *Sellers et al. (1996)* |
| Surface layer drag coefficient (opt_sfc) | Original Noah *Chen et al. (1997)* |
| Soil moisture for stomatal resistance (opt_btr) | Noah (soil moisture) |
| Runoff (opt_run) | TOPMODEL with groundwater *Niu et al. (2007)* |
| Supercooled liquid water (opt_frz) | no iteration *Niu and Yang (2004)* |
| Soil permeability (opt_inf) | linear effects, more permeable *yue Niu and liang Yang (2006)* |
| Radiative transfer (opt_rad) | modified two-stream |
| | (gap = F(solar angle, 3D structure ...)<1-FVEG) |
| | *Yang and Friedl (2003)*, *Niu and Yang (2004)* |
| Ground surface albedo (opt_alb) | BATS *Yang Z.-L. and Vinnikov. (1997)* |
| Precipitation (snow/rain) partitioning (opt_snf) | *Jordan (1991)* |
| Soil temperature lower boundary (opt_tbot) | TBOT at ZBOT (8m) read from a file (original Noah) |
| Soil/snow temperature time scheme (opt_sfc) | semi-implicit; flux top boundary condition |
| Surface resistance to evaporation/sublimation (opt_rsf) | *Sakaguchi and Zeng (2009)* |
| Glacier treatment (opt_gla) | include phase change of ice |

# Appendix A

Following *Monaghan et al. (2018)*, NOAH-MP parameter file MPTABLE.TBL has been modified, and it can be used for simulations over Alaska. These modifications improved the model's capability to capture cold surface temperature and meteorological profiles (e.g. wind speed, relative humidity, temperature) over Alaska.

**Appendix B**

Fuentes size-resolved sea-spray source flux





$$\frac{dF_{\rm o}}{dlogD_{\rm po}} = \frac{dF_{\rm p}}{dlogD_{\rm po}} \times {\rm W} = \frac{Q}{A_{\rm b}} \times \frac{dN_{\rm T}}{dlogD_{\rm po}} \times {\rm W} \tag{B1}$$

where W(U) is Monahan and O'Muircheartaigh whitecap coverage, $dF_p/dlogD_{p0}$ is the size-resolved particle flux per unit time and water surface covered by bubbles, $D_{p0}$ is the dry diameters, Q is the sweep air flow, $A_b$ is the total surface area covered by bubbles, $dNT/dlogD_{p0}$ is the particle size distribution (the sum of four log-normal modes) and is equal to:

$$\frac{dN_{\rm T}}{dlogD_{\rm po}} = \sum_{i=1}^{4} \frac{dN_{\rm T,i}}{dlogD_{\rm po}} = \sum_{i=1}^{4} \frac{N_{\rm T,i}}{\sqrt{2\pi} \times \log\sigma_{\rm i}} \times exp[-\frac{1}{2} \times (\frac{log\frac{D_{\rm po}}{D_{\rm pog,i}}}{log\sigma_{\rm i}})^2] \tag{B2}$$

where i is the sub-index for the mode number and $N_i$, $D_{p0g,i}$ and $\sigma_i$ are the total particle number, geometric mean and geometric standard deviation for each log-normal mode. $N_{T,i}$ and $D_{p0g,i}$ are depending on parameters $a_i$ and $\beta_i$ derived from polynomial and exponential regressions defining the total particle number and geometric mean diameter of the log-normal modes, and can be found in Table 5 *Fuentes et al. (2010)*.

**Appendix C**

In this APPENDIX, the biases and RMSEs are calculated for each site, as shown in Fig. 1, and are shown in the tables below. Each table corresponds to a site and for the available observed aerosol concentrations, such as $Na^+$, $Cl^-$, $SO_4^{2-}$ (total and non-sea component), $NO_3^-$, $NH_4^+$ and OC. Bias is calculated as the difference between model simulation and observation.





**Table C1.** Biases and RMSEs, in $\mu$ g m$^{-3}$, are calculated for aerosols at the Alert, Canada, during January and February 2014 and for CONTROL and HEM_NEW simulations at 100km.

|  | CONTROL | | HEM_NEW | |
| --- | --- | --- | --- | --- |
|  | Bias | RMSE | Bias | RMSE |
| Na$^+$ | 0.81 | 0.91 | 0.12 | 0.18 |
| Cl$^-$ | 1.05 | 1.2 | -0.03 | 0.19 |
| NO$_3^-$ | 0.28 | 0.30 | 0.25 | 0.22 |
| NH$_4^+$ | -0.003 | 0.01072 | 0.007 | 0.01079 |
| SO$_4^{2-}$ | 0.06 | 0.1 | 0.02 | 0.11 |

**Table C2.** Biases and RMSEs, in $\mu$ g m$^{-3}$, are calculated for aerosols at Villum Research station, Greenland, during January and February 2014 and for CONTROL and HEM_NEW simulations at 100km.

|  | CONTROL | | HEM_NEW | |
| --- | --- | --- | --- | --- |
|  | Bias | RMSE | Bias | RMSE |
| Na$^+$ | 1.3 | 1.4 | 0.25 | 0.26 |
| Cl$^-$ | 1.9 | 2.1 | 0.22 | 0.24 |
| NO$_3^-$ | 0.25 | 0.26 | 0.17 | 0.19 |
| NH$_4^+$ | -0.001 | 0.006 | 0.01 | 0.01 |
| SO$_4^{2-}$ | 0.05 | 0.1 | 0.08 | 0.1 |


**Table C3.** Biases and RMSEs, in $\mu$ g m$^{-3}$, are calculated for aerosols at Zeppelin, Norway, during January and February 2014 and for CONTROL and HEM_NEW simulations at 100km.

|  | CONTROL | | HEM_NEW | |
| --- | --- | --- | --- | --- |
|  | Bias | RMSE | Bias | RMSE |
| Na$^+$ | 3.31 | 4.4 | 0.36 | 0.78 |
| Cl$^-$ | 4.86 | 6.48 | 0.22 | 0.73 |
| NO$_3^-$ | 0.13 | 0.36 | 0.01 | 0.29 |
| NH$_4^+$ | -0.03 | 0.077 | -0.02 | 0.076 |
| SO$_4^{2-}$ | 0.16 | 0.25 | 0.32 | 0.45 |





**Table C4.** Biases and RMSEs, in $\mu$ g m$^{-3}$, are calculated for aerosols at Simeonof, south of Alaska, during January and February 2014 and for CONTROL and HEM_NEW simulations at 100km.

| | CONTROL | | HEM_NEW | |
|---|---|---|---|---|
| | Bias | RMSE | Bias | RMSE |
| $Na^+$ | 1.4 | 2.5 | 0.3 | 0.6 |
| $Cl^-$ | 2.0 | 3.7 | 0.1 | 0.7 |
| $NO_3^-$ | 0.12 | 0.23 | 0.08 | 0.20 |
| $SO_4^{2-}$ | -0.2 | 0.25 | -0.05 | 0.26 |
| OA | -0.08 | 0.1 | -0.05 | 0.08 |

**Table C5.** Biases and RMSEs, in $\mu$ g m$^{-3}$, are calculated for aerosols at Gates of the Arctic, south of Alaska, during January and February 2014 and for CONTROL and HEM_NEW simulations at 100km.

| | CONTROL | | HEM_NEW | |
|---|---|---|---|---|
| | Bias | RMSE | Bias | RMSE |
| $Na^+$ | 0.6 | 0.9 | 0.2 | 0.3 |
| $Cl^-$ | 0.7 | 1.2 | 0.1 | 0.3 |
| $NO_3^-$ | 0.3 | 0.3 | 0.2 | 0.2 |
| $SO_4^{2-}$ | -0.04 | 0.2 | -0.1 | 0.2 |
| OA | -0.24 | 0.28 | -0.21 | 0.26 |

**Table C6.** Biases and RMSEs, in $\mu$ g m$^{-3}$, are calculated for super-micron aerosols at Utqiaġvik, north of Alaska, during January and February 2014 and CONTROL and HEM_NEW simulations at 100km.

| | CONTROL | | HEM_NEW | |
|---|---|---|---|---|
| | Bias | RMSE | Bias | RMSE |
| $Na^+$ | 0.3 | 0.37 | -0.07 | 0.25 |
| $Cl^-$ | 0.27 | 0.48 | -0.26 | 0.51 |
| $NO_3^-$ | 0.26 | 0.3 | 0.13 | 0.17 |
| $NH4_4^+$ | -0.0004 | 0.00368 | -0.001 | 0.0037 |
| $SO4_4^{2-}$ | 0.005 | 0.005 | -0.01 | 0.06 |



**Table C7.** Biases and RMSEs, in $\mu$ g m$^{-3}$, are calculated for sub-micron aerosols at Utqiaġvik, north of Alaska, during January and February 2014 and CONTROL and HEM_NEW simulations at 100km.

|  | CONTROL | | HEM_NEW | |
|---|---|---|---|---|
|  | **Bias** | **RMSE** | **Bias** | **RMSE** |
| **Na$^+$** | -0.485 | 0.66 | -0.489 | 0.67 |
| **Cl$^-$** | -0.116 | 0.361 | -0.124 | 0.364 |
| **NO$_3^-$** | -0.065 | 0.162 | -0.054 | 0.158 |
| **NH$_4^+$** | -0.069 | 0.106 | -0.057 | 0.100 |
| **SO$_4^{2-}$** | -0.621 | 0.875 | -0.591 | 0.853 |

**Table D1.** Biases and RMSEs, in $\mu$ g m$^{-3}$, are calculated between ALASKA_NEW_JAN, ALASKA_NEW_FEB and in-situ meteorological parameters derived from NOAA Baseline Observatories during the campaign's periods in January and February 2014. Bias was calculated as the difference between model simulation and observations.

|  | January campaign | | February campaign | |
|---|---|---|---|---|
|  | Bias | RMSE | Bias | RMSE |
| 2m Temperature | 0.1 | 1.9 | -1.0 | 3.2 |
| 10m Temperature | -0.03 | 1.8 | -0.66 | 2.7 |
| 10m Wind speed | 0.08 | 1,4 | -0.33 | 1.7 |
| 10m Wind direction | -11.2 | 13.2 | -11.2 | 39.0 |

## Appendix D

Surface observations are used to validate the meteorological conditions that occur over Utqiaġvik and Alaska in wintertime. See also discussion in sub-section 5.1 in the main text. The model is validated against the surface (hourly) observations obtained from National Oceanic and Atmospheric Administration / Earth System Research Laboratory / Global Monitoring Division 800 (NOAA/ESRL/GMD) Baseline Observatories. Also, radiosondes data are used to evaluate the model's performance at different altitudes. Radiosonde data (every 12h) are derived from Integrated Global Radiosonde Archive version 2 (IGRA 2) (*Durre et al. (2018)*). Site is located at latitude: 71.2889 and longitude: -156.7833.





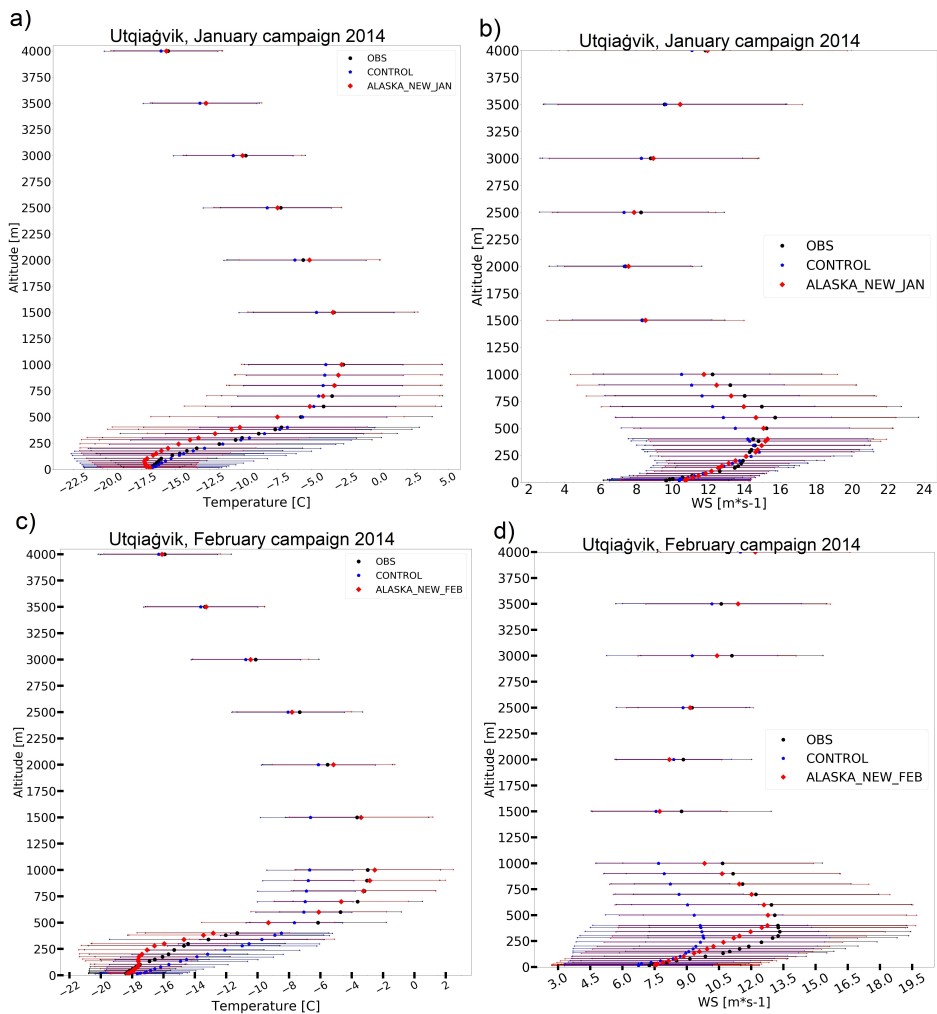

**Figure D1.** Average temperatures, in degrees C, and wind speeds, in ms$^{-1}$, as a function of altitude (m), up to 4km, during (a,b) January and (c,d) February campaign in 2014, at Utqiaġvik, Alaska. The observations are shown in black (cicle). The blue pentagon shows the model results for the CONTROL simulation (at 100km) and the red diamond shows the model results for the NEW_ALASKA_JAN and NEW_ALASKA_FEB simulation. Observations are derived from IGRA2 and are available every 12h (0Z and 12Z, UTC). For the comparison, model output at 0 and 12Z UTC are used. The corresponding horizontal lines show the standard deviation.





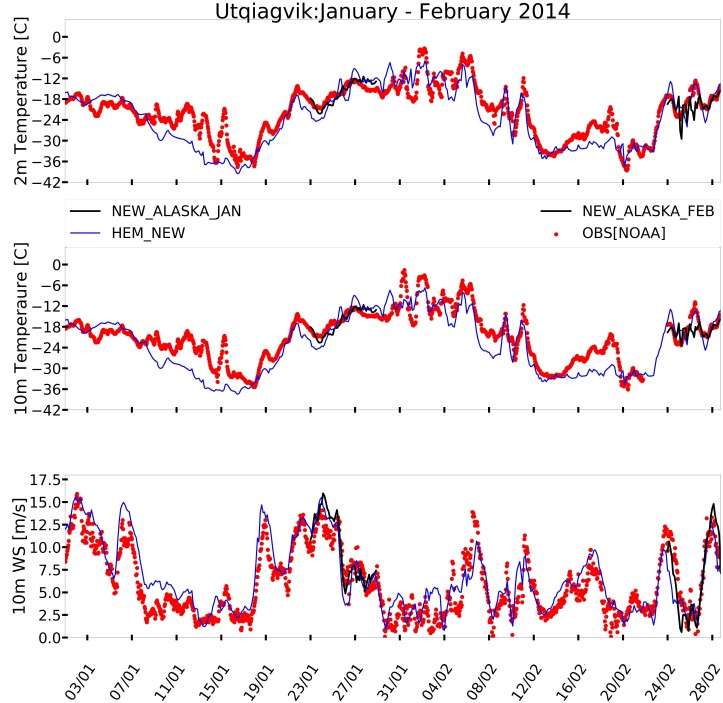

**Figure D2.** Time series of observed and modelled 2m and 10m temperature, and 10m wind speed, at Utqiaġvik, Alaska, in UTC. The observations are shown in red and derived from the NOAA observatory. The blue line shows the results for the HEM_NEW simulation at 100km, while the black line shows the results for ALASKA_NEW_JAN and ALASKA_NEW_FEB simulations at 20km. The observations are hourly, while the model output is every 3h.

**Appendix E**

Here the bias and RMSE are shown between ALASKA_NEW_JAN and ALASKA_NEW_FEB and the observations for
Utqiaġvik at 20km.

**Appendix F**

This APPENDIX shows the comparison for the Gates of the Arctic site at 20 km, for ALASKA_NEW_FEB and ALASKA_CONTROL_FEB. The observations are only available for the February campaign, daily averaged in local Alaskan time every three days.

Also, the table below shows biases and RMSEs, in $\mu$ g m$^{-3}$, for all available aerosol species at the Gates of the Arctic.





**Table E1.** Biases and RMSEs, in $\mu$ g m$^{-3}$, are calculated for aerosols at Utqiaġvik, north of Alaska, during January 2014 and for ALASKA_CONTROL_JAN and ALASKA_NEW_JAN simulations at 20km.

|  | ALASKA_CONTROL_JAN | | ALASKA_NEW_JAN | |
|---|---|---|---|---|
|  | Bias | RMSE | Bias | RMSE |
| $Na^+$ | -0.31 | 0.38 | -0.16 | 0.26 |
| $Cl^-$ | -0.50 | 0.59 | -0.33 | 0.43 |
| $NO_3^-$ | -0.040 | 0.07 | 0.039 | 0.09 |
| $SO_4^{2-}$ | -0.396 | 0.414 | -0.398 | 0.417 |
| $NH_4^+$ | -0.033 | 0.038 | -0.035 | 0.040 |

**Table E2.** Biases and RMSEs, in $\mu$ g m$^{-3}$, calculated for aerosols at Utqiaġvik, north of Alaska, during February 2014 and for ALASKA_CONTROL_FEB and ALASKA_NEW_FEB simulations at 20km.

|  | ALASKA_CONTROL_FEB | | ALASKA_NEW_FEB | |
|---|---|---|---|---|
|  | Bias | RMSE | Bias | RMSE |
| $Na^+$ | -1.29 | 1.40 | -1.18 | 1.30 |
| $Cl^-$ | -1.90 | 1.92 | -1.78 | 1.80 |
| $NO_3^-$ | -0.20 | 0.40 | -0.11 | 0.38 |
| $SO_4^{2-}$ | -1.019 | 1.322 | -1.020 | 1.326 |
| $NH_4^+$ | -0.045 | 0.097 | -0.043 | 0.10 |

**Table F1.** Biases and RMSEs, in $\mu$ g m$^{-3}$, are calculated for aerosols at Gates of the Arctic, north of Alaska, during February campaign and for ALASKA_CONTROL_FEB and ALASKA_NEW_FEB simulations at 20km.

|  | ALASKA_CONTROL_FEB | | ALASKA_NEW_FEB | |
|---|---|---|---|---|
|  | Bias | RMSE | Bias | RMSE |
| $Na^+$ | 0.344 | 0.346 | 0.342 | 0.341 |
| $Cl^-$ | 0.155 | 0.1578 | 0.154 | 0.1573 |
| $NO_3^-$ | 0.47 | 0.48 | 0.42 | 0.47 |
| $SO_4^{2-}$ | 0.24 | 0.24 | 0.23 | 0.23 |
| OA | -0.198 | 0.2446 | -0.197 | 0.2445 |





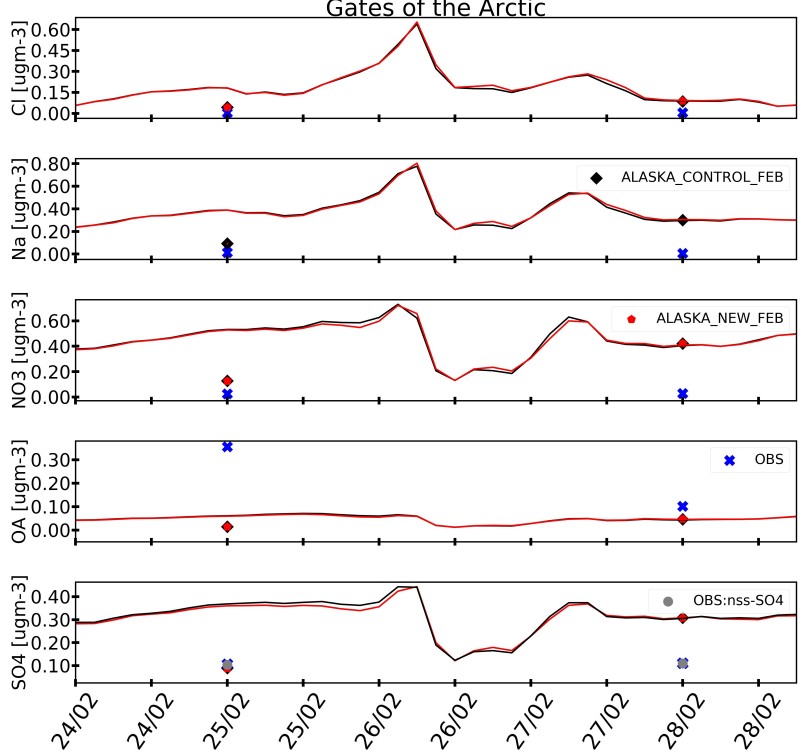

**Figure F1.** Model inter-variability during February campaign. Model simulations are validated against aerosols at the gates of the Arctic site, north of Alaska. The black line shows ALASKA_CONTROL_FEB simulation and the black symbol the daily averaged values. The red line shows ALASKA_NEW_FEB simulation and the red pentagon the daily averaged values. The blue star indicates averaged daily observations. Observations and model are in local Alaskan time. Observed and modelled $SO_4^{2-}$ is total $SO_4^{2-}$.

*Author contributions.* The first author (EI) implemented the updates, performed the simulations and the analysis, and drafted the paper. KSL designed the study and contributed to the interpretation of results and the analysis, and to the writing of the paper. JCR, LM and TO contributed to discussions about the model setup and simulations. JCR, LM, KP, RMK, PKQ and LU contributed to the analysis and interpretation of the results. AM and HS contributed to the interpretation of the results. All co-authors contributed to the paper and the discussions about the results.

*Competing interests.* The authors declare that they have no conflict of interest.

*Acknowledgements.* This study is supported by the PhD programme of the Make our planet great again (MOGPA) French initiative. We also acknowledge support from French ANR CASPA (Climate-relevant Aerosols and Sources in the Arctic). Computer modeling performed by Eleftherios Ioannidis, benefited from access to IDRIS HPC resources (GENCI allocations A009017141 and A011017141) and



the IPSL mesoscale computing center (CICLAD: Calcul Intensif pour le Climat, l'Atmosphère et la Dynamique). We acknowledge the use of MOZART-4 global model output available at http://www.acom.ucar.edu/wrf-chem/mozart.shtml. We acknowledge use of the WRF-Chem preprocessor tool mozbc provided by the Atmospheric Chemistry Observations and Modeling Lab (ACOM) of NCAR. We would like to thank Arctic Monitoring & Assessment Programme (AMAP) and Zbigniew Klimont for ECLIPSE v6b anthropogenic emissions. K.A.P. acknowledges funding from the US National Science

Foundation (OPP-1724585). This is PMEL contribution number 5366. Also, we would like to thank the IMPROVE database for the aerosol observations in Alaska. IMPROVE is a collaborative association of state, tribal, and federal agencies, and international partners. US Environmental Protection Agency is the primary funding source, with contracting and research support from the National Park Service. The Air Quality Group at the University of California, Davis is the central analytical laboratory, with ion analysis provided by Research Triangle Institute, and carbon analysis provided by Desert Research Institute. Meteorological data provided by NOAA Global Monitoring Laboratory,

Boulder, Colorado, USA (https://esrl.noaa.gov/). We would like to thank Josh Jones for providing us with detailed maps of sea-ice fraction plots from the radar at Utqiaġvik. The Villum Foundation is gratefully acknowledged for financing the establishment of the Villum Research Station. Finally, we would like to thank Aas Wenche (Norwegian Institute for Air Research) and Sharma Sangeeta (Environment and Climate Change Canada, Science and Technology Branch, Toronto, Canada) for providing us with EBAS observations at Zeppelin and Alert, respectively. Authors also want to thank Canadian Forces Services, Alert, NU for maintaining the site. The observations at Zeppelin have

been supported by the Norwegian Environment Agency (grant no. 17078061).





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
