# Peer review of "Modelling wintertime sea-spray aerosols under Arctic Haze conditions"

_EGUsphere, 2022_

## Author Response (AR1)

We would like to thank both reviewers for their positive feedback and valuable comments. We have revised the manuscript accordingly. Please find our responses to reviewer comments below in blue and the revised text in red. The reviewers' comments are in black. The revised manuscript with and without the changes highlighted is also provided. First, we summarise the main revisions to the manuscript. At the end of each reviewer's reply we provide a list with the references we added in the revised manuscript.

Based on the major comment from Rev. 1, we have revised the text on the model evaluation of Arctic SO4  $^{2-}$  including sub-micron nss- SO4  $^{2-}$  and discuss our findings within the context of other studies. We also replied to all Rev. 1's minor comments.

To address the comments and concerns made by Rev. 2, the paper has been re-organised and shortened. In particular, we revised the Title and re-wrote the Abstract, Introduction, the section on evaluation of the 100km results (old Section 4, now Section 5) and the Conclusions. The SSA emission scheme used in the base version of the model, together with the updates to this scheme, are now described in a separate Section 4.

We also would like to inform the reviewers that we found an error in the model simulations. Due to the complexities of the WRF-Chem model, unfortunately, the flag activating dinitrogen pentoxide ( $N_2O_5$ ) hydrolysis was switched off. We have rerun all the simulations with N2O5 hydrolysis on, and the corrected results are included in the revised manuscript. Since N2O5 hydrolysis is an important source of nitric acid (HNO3) at night-time, we assessed the impact on our results. Whilst changes in HNO3, and thus on NO3- aerosols, can be important over NOx emission regions, we found a rather small influence on modelled NO3- in the Arctic (maximum changes of up to 0.1  $\mu$ m-3 for super-micron and sub-micron NO3-). It appears that since N2O5 hydrolysis is faster where water vapour (humidity) is higher, the effects are larger over more southerly mid-latitude emission regions. Since humidity over the Arctic in winter is low, the results are affected to a lesser extent over this region. We include figures to illustrate this below. Figure 1 shows the absolute changes for sub-micron NO3- during February 2014 over the Arctic region. Note that the only difference between the two simulations for HEM\_NEW is N2O5 hydrolysis. The rest of the set-up is the same as described in the submitted manuscript. Figure 2 illustrates the differences in modelled inorganic aerosols between model runs (HEM NEW) with and without N2O5 hydrolysis compared to observations at Zeppelin. Only small differences are found at this and other Arctic sites.

Figure 1: Average absolute differences in sub-micron  $NO_3^-$  concentrations (in µgm-3) between HEM\_NEW (with N2O5 hydrolysis) and HEM\_NEW (without N2O5 hydrolysis) runs during February 2014 at the surface. The black star in northern Alaska shows where Utqiaġvik is located. The black circle shows Alert, Canada, the black diamond shows Villum in Greenland, while the black pentagon shows Zeppelin, Svalbard. All the results are shown north of 50N.

---

## Author Response (AR2)

*We thank the reviewer for their remarks. Below we include reviewer's comments in black and our replies in blue. We also include text from the submitted paper and the revised version in red. Please note that two co-authors have been added to the paper, since their team is responsible for the Utqiaġvik SMPS data we included in the revised manuscript.*

Reviewer 1

The paper has been significantly improved. If my three additional comments are properly addressed, I would recommend publication. Comments 1 and 2 are the reasons I rate scientific quality as fair.

1) The variations in the underestimations of Na+, Cl-, and SO4= look to be linked in Figure 5b (submicron). This and the broad open region in Figure 3 suggest the model is not transporting enough of these components to Northern Alaska. I think that emphasizing the argument that the underestimation is a regional issue (lines 406-410) and diminishing the argument that it is due to an underestimation in transport from mid latitudes (line 412) is not supported by Figures 3 and 5b, and it does not help your model. In your conclusions on lines 713-715, you don't even mention latitudinal transport, choosing to put the blame on local sources and heterogeneous reactions, which I find there to be much less support for.

*While we agree with the reviewer about the importance of long-range transport, as well as local sources, we feel that their point is not entirely true. We discuss the underestimation of sub-micron SSA at Utqiaġvik (lines 365-370) clearly stating that this could be due to issues with long-range transport or wet removal. We also note (lines 400-401) that discrepancies in modelled nss-SO$_4^{2-}$ at Zeppelin could be due to uncertainties in simulating the vertical distribution or transport from Eurasian source regions. In lines 406-410, we mention that model underestimation of nss-SO$_4^{2-}$ at Utqiaġvik may be due to long-range transport as well as local sources, an argument supported by the study of Kirpes et al. (2018). However, we rephrased the text to be clearer.*

*Old text: As noted by KRP18 and KRP19, this is likely to be due to the local influence from the North Slope of Alaska (NSA) oil fields to the east. In a companion paper, Ioannidis et al. (2022, in prep.) the influence of these regional emissions on BC at Barrow is investigated during winter 2014 finding that up to 30-50% of BC may originate from this source. (Lines 401-404)*

*Revised text: As noted by KRP18 and KRP19, there is a local influence from the North Slope of Alaska (NSA) oil fields to the east, and these emissions may be underestimated in the model. In a companion paper investigating BC at Utqiaġvik, it is estimated that up to 30-50% of BC originates from these regional emissions (Ioannidis et al. 2023 in prep.). (Lines 401-404)*

*Old text: As well as local sources, model difficulties simulating sub-micron nss-SO$_4^{2-}$ at Utqiaġvik may be due to underestimation in transport of mid-latitude sources. SO$_4^{2-}$ formation mechanisms under dark, cold winter conditions may also be lacking in the model. (Lines 411-413)*

*Revised text: In addition to local sources, difficulties simulating sub-micron nss-SO$_4^{2-}$ at Utqiaġvik may be due to an underestimation in the transport of nss-SO$_4^{2-}$ from mid-latitudes to the Arctic or issues related to deposition (as noted earlier for SSA), as also discussed in previous studies such as Whaley et al. (2022). SO$_4^{2-}$ formation mechanisms under dark, cold winter conditions may also be lacking in the model. (Lines 406-410)*

*In the Conclusions, while we discuss local sources and heterogeneous reactions (lines 713-715), we also state (lines 717-718) that uncertainties in transport and removal processes may be contributing to model discrepancies in wintertime aerosols. This statement covers all aerosols, including SO$_4^{2-}$ at remote Arctic sites.*

*Old text: Uncertainties in model transport and wet and dry deposition processes may also be responsible for deficiencies in modelled wintertime Arctic aerosols (Whaley et al., 2022). (Lines 716 - 717)*

*Revised text: The above, combined with uncertainties in model transport and wet and dry deposition processes contribute to model deficiencies in simulating wintertime Arctic aerosols (Whaley et al., 2022). (Lines 717-718)*

*Overall, we believe that model performance is dependent on many factors, including local/regional processes/sources and long-range transport.*

2) Figure 11 - I don't know where the observed distributions for January and February came from, but the January and February distributions for Barrow in Freud et al. do not show an Aitken mode higher than the accumulation mode. In fact, they don't show much of an Aitken mode at all, which is common for the Arctic Haze. The observed distributions in Figure 11 look like summertime distributions. Please check and correct.

*We thank the reviewer for these remarks. We agree that the distributions in Figure 11 in the revised manuscript have similarities with distributions for July, August, as shown in Figure 4 in Freud et al. (2017). Their figure shows monthly mean size distributions averaged over different years (2007-2009, 2013-2015). Freud et al. (2017) also show (Figure 2) that not a lot of data is available for Utqiaġvik during 2014, and especially during January. In our paper, Figure 11 shows distributions for shorter periods (end of January and February 2014). We obtained our data from [ftp://ftp.cmdl.noaa.gov/aerosol/brw/smps/](ftp://ftp.cmdl.noaa.gov/aerosol/brw/smps/). The data were processed by colleagues in TROPOS in Germany who are responsible for these measurements at Utqiaġvik.*

*To check the SPMS data used in Figure 11, we compared to the SMPS data from [https://doi.org/10.1594/PANGAEA.877333](https://doi.org/10.1594/PANGAEA.877333) as referred to in Freud et al. (2017) and plotted average observed size distributions during the January and February 2014 campaign periods (see Figs 1 and 2 below). There are some differences between the TROPOS SMPS data and the homogenized hourly average SMPS data from Freud et al. (2017), especially during February. We believe these differences are due to filtering, based on wind direction criteria, applied by Freud et al. (2017) to exclude influences from the NSA (Prudhoe Bay) oil fields (see section 2.2 in Freud et al. 2017), and especially local Utqiaġvik village pollution (fresh small particles). Freud et al. (2017) state: "For Barrow, all recorded aerosol data are normally dismissed when wind directions are between 130 and 360 or when the winds are weaker than $0.5ms^{-1}$. However, we have noticed that despite this, there were still indications of local pollution both in the SMPS and Nephelometer data just after a local wind shift from the potentially "polluted" to the "clean" sector."*

*Freud et al. (2017) also applied extra filtering:*

*"An extra prerequisite was thus added, requiring that the winds from the clean sector need to be persistent for at least 24 h. In addition, data were omitted during the first 2 h after the wind shift. This procedure does not necessarily filter observations that are potentially affected by the large oil and gas extraction fields at Prudhoe Bay 300 km to the east-southeast (Kolesar et al., 2017), but this should not be regarded as local pollution anyway."*

*In the study by Kolesar et al. (2017), including analysis of SMPS data, and including winter 2014, it is shown that there are particle-growth events at Utqiaġvik during winter (January and February). For example, the authors refer to a particle-growth event when the air masses originate from Prudhoe*

*Bay, associated with high concentrations of volatile organic compounds (VOCs) which may form secondary organic aerosols.*

*Since the SMPS data we are using is not filtered for wind direction this can explain the differences between Figure 11 in our manuscript and Freud et al. (2017). Thus, based on the above points, we do not think that there is a problem with the SMPS data used in Figure 11. In the revised manuscript we added the following text regarding the SMPS data:*

No wind speed criteria have been applied to exclude local (Utqiaġvik town) or regional (e.g. North Slope of Alaska oil fields) pollution. *(Lines 200-201)*

Freud et al. (2017) reported similar wintertime magnitudes in the accumulation mode (diameter range 100-150 nm) at Utqiaġvik, averaging between $1 \times 10^2$ and $2 \times 10^2$ particles per $cm^3$, whereas smaller number concentrations were reported for particles less than 50nm than shown here. These differences can be explained by the fact that the SMPS data used here does not exclude local/regional pollution based on wind speed criteria unlike Freud et al. (2017). Local or regional pollution influenced by Utqiaġvik town or NSA oil fields could lead to new particle growth events (small particles) in the absence of sunlight, as discussed by Kolesar et al. (2017). Freud et al. (2017) also noted that particle number concentrations are higher at Utqiaġvik and Tiksi (sites in proximity to local sources) compared to other Arctic sites (Alert, Villum and Zeppelin). *(Lines 599-605)*

[Figure]

*Figure 1: Average observed size distributions for 24-28 January 2014 at Utqiaġvik. The blue line and dots show the TROPOS SMPS data (initially every 5', then hourly averaged and then averaged over 24-28 January), while the red line and dots show the average (initially hourly data and then averaged over 24-28 January) and filtered data used in Freud et al. (2017).*

[Figure]

*Figure 2: Average observed size distributions: same as Fig. 1 but for 24-28 February 2014 at Utqiaġvik.*

3) In Figure 2, The diamond is Alert and the circle is Villium. *Thank you. Corrected*